# MOUCHI: MITIGATING OVER-FORGETTING IN UN-LEARNING COPYRIGHTED INFORMATION

## ABSTRACT

Large language models are trained on massive internet datasets, which may inadvertently memorize illegal copyrighted content, making its inclusion unavoidable. Unlearning is a potential solution to remove such content. However, existing unlearning methods often suffer from *over-forgetting*, where the process unintentionally erases knowledge similar to the copyrighted content that falls under fair use and should be preserved. To address this issue, we propose MOUCHI, a novel unlearning framework that introduces the concept of *derivative knowledge*, a subset of information derived from copyrighted content that must be retained during unlearning. MOUCHI first generates derivative knowledge and then incorporates a derivative loss function into the unlearning process to mitigate over-forgetting in unlearning copyrighted content. Due to its plug-and-play nature, MOUCHI can be effortlessly integrated into existing unlearning methods. Experimental results show that MOUCHI reduces unintended knowledge loss, improving performance by **up to 145%** compared to baseline methods when evaluated on the derivative set.

## 1 INTRODUCTION

Large language models (LLM) have grown exponentially in scale and sophistication, driven by increasing parameter sizes and extensive training datasets. However, this reliance on vast amounts of data introduces a significant challenge of inadvertently including copyrighted material, often propagated illegally online. Copyright concerns in LLMs have become especially prominent due to their widespread use, making these models more vulnerable to misuse (Carlini et al., 2021; Hernandez et al., 2022; Chang et al., 2023). Karamolegkou et al. (2023) observe a linear correlation between the size of a language model and its tendency to generate verbatim copies of famous books, which constitutes a clear violation of copyright law. Moreover, new regulations, such as the EU AI Act (Friedl & Gasiola, 2024), mandate that all general-purpose AI systems, including LLMs, adhere to union copyright laws and the general data protection regulation, granting rights holders the ability to protect their works from unauthorized text and data mining and ensuring the *right to be forgotten* (Hoofnagle et al., 2019). Consequently, there is a growing need for efficient methods to remove copyrighted data from LLMs upon request (Ren et al., 2024; Wei et al., 2024).

Machine unlearning has emerged as a novel approach to address this issue. By applying the principles of unlearning, it is possible to systematically remove specific information from a trained model, thereby mitigating the concern of copyright infringement within LLMs. Recent papers (Yao et al., 2023; Eldan & Russinovich, 2023; Jang et al., 2022) use machine unlearning to remove harmful content (e.g., personal information, toxic passage, and copyrighted information) from LLMs by employing variants of gradient ascent (GA) during the unlearning process. GA adjusts the model parameters to maximize the loss on specific data points being unlearned, thus diminishing the model's ability to generate outputs related to the undesired data. However, these techniques have notable shortcomings, particularly the problem of over-forgetting, which we introduce in this paper.

*Over-forgetting* refers to an unintentional erasure of more than just the targeted data. It is a critical issue that occurs when we cannot fully control the GA process. In tasks focused on removing toxic passages or private information, over-forgetting is less problematic since the goal is complete removal. In contrast, when the target data involves copyrighted content, over-forgetting becomes more problematic, as it not only erases the targeted copyrighted material but also leads to the unintended loss of relevant knowledge that falls within the bounds of fair use. We illustrate this problem in Figure

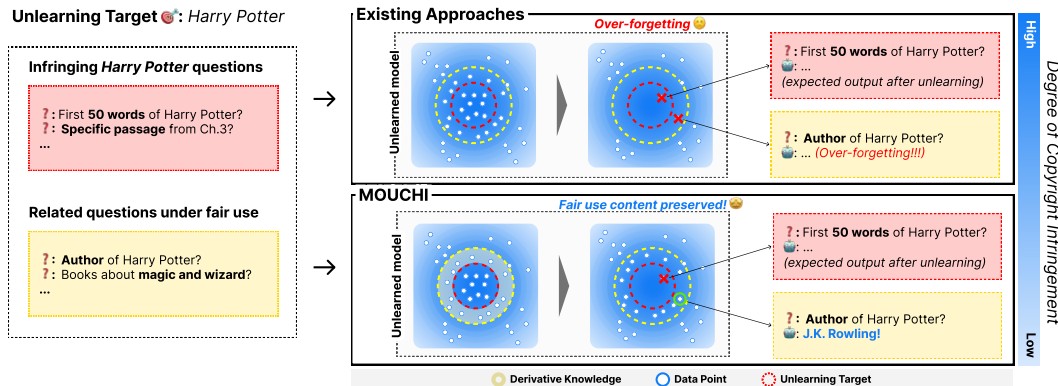

Figure 1: Over-forgetting problem in existing approaches, highlighted by the absence of dots within the red and yellow boundaries in the embedding space and how MOUCHI solves the problem by maintaining the questions within this region.

1, where existing approaches fail to retain non-infringing information, such as identifying the author of a book or answering general questions about related themes. If over-forgetting consistently occurs across different unlearning targets, the overall performance of the LLM will be severely degraded.

To address the issue of over-forgetting, this paper proposes a novel framework for **M**itigating **O**ver-forgetting in **U**nlearning **C**opyrig**H**ted **I**nformation within LLMs, **MOUCHI**, which identifies the problem of over-forgetting in existing methods and introduces the concept of ***derivative knowledge***—a subset derived from the target copyrighted information that must be retained during unlearning. This subset consists of knowledge that falls within fair use boundaries and must be preserved during the unlearning process to prevent unintentional loss. Our MOUCHI framework generates derivative knowledge from the target information, using KL divergence as a metric to assess semantic closeness. By doing so, we enable the model to identify the derivative set while distinguishing it from the target forget set. As in the lower part of Figure 1, MOUCHI preserves the fair use content (the dots within the red and yellow boundaries), allowing the model to retain its ability to answer general or related questions while still removing the infringing content. It then incorporates the derivative loss into the unlearning process, ensuring that the derivative knowledge is preserved while the copyrighted content is removed. Moreover, MOUCHI is model-agnostic and can be seamlessly integrated into existing unlearning methods, providing enhanced control over the unlearning process without compromising performance. By virtue of the derivative knowledge, MOUCHI effectively maintains the model's overall capabilities while mitigating over-forgetting. Our **contributions** are as follows.

- We analyze and identify the over-forgetting problem in current LLM unlearning methods.
- To address the over-forgetting in LLM unlearning, we propose MOUCHI, a new unlearning framework that generates and incorporates a set of derivative knowledge. A derivative loss function is also introduced to realize the unlearning process under our framework.
- We demonstrate that MOUCHI can be seamlessly integrated into existing unlearning methods, maintaining performance comparable to traditional approaches while providing better control over over-forgetting. The experimental results show that the models with MOUCHI exhibit up to 145% higher utility on the derivative set compared to the baselines.

## 2 RELATED WORK

**LLM Unlearning** Traditional machine unlearning techniques, which aim to remove specific knowledge from models without full retraining, face challenges when applied to LLMs due to scalability issues and the decentralized nature of their training data (Liu et al., 2024b). §A reviews general machine unlearning techniques and their limitations. In response to these challenges, recent research has proposed scalable and effective unlearning approaches tailored specifically for LLMs. Lu et al. (2022); Jang et al. (2022); Yao et al. (2023) explore various fine-tuning methods, including reward-reinforced model fine-tuning and gradient ascent-based fine-tuning, to unlearn specific content from LLMs. Other methods (Wang et al., 2023; Yu et al., 2023; Chen & Yang, 2023) involve techniques like KL-divergence-based fine-tuning, weight importance-informed relabeling, and parameter-efficient

task vector tuning. Lately, a study (Zhang et al., 2024) introduces negative preference optimization to address the problem of catastrophic collapse caused by the lack of control over gradient ascent variants at higher temperatures. Despite these advancements, most methods focus only on general unlearning in LLMs, overlooking the specific issue of unlearning copyrighted information.

**Unlearning Copyrighted Information**    Research on the unlearning of copyrighted content from LLMs remains relatively underexplored. This unlearning requires distinct treatment from general unlearning due to a heightened susceptibility to over-forgetting. Li et al. (2024); Henderson et al. (2023) discuss the potential risks of copyright infringement in LLMs and how to detect the copyright content from the output. However, they do not focus on the method of removal. Eldan & Russinovich (2023); Chen et al. (2024) explore unlearning for specific books like Harry Potter and The Lord of the Rings. However, these approaches lack scalability as they are demonstrated only in a specific book. They also fail to observe the differences between copyrighted content unlearning and general unlearning. To address these limitations, Dou et al. (2024) propose a method that sequentially removes copyrighted content from multiple books; meanwhile, a few papers introduce novel datasets for copyright removal, along with baselines for unlearning tasks (Maini et al., 2024; Liu et al., 2024c). In addition, recent studies (Liu et al., 2024a; Dou et al., 2024) mention an open problem of unintended knowledge loss, particularly the erasure of fair-use content during the unlearning process. As a solution, MOUCHI effectively mitigates this issue by introducing derivative knowledge that falls within the bounds of fair use.

## 3    PRELIMINARIES

Let $\mathcal{D} = \{(x_i, y_i)\}_{i=1}^N$ is a training dataset for an LLM having $N$ input-output pairs, where $x_i$ and $y_i$ represent the input and its corresponding output, respectively. In practice, $x_i$ and $y_i$ usually represent a question and answer (QnA) pair. We focus on QnA pairs because QnA is the primary way that users interact with LLMs in real-world scenarios.

**Definition 3.1** (FORGET SET).  Given a training dataset $\mathcal{D}$, a *forget set* is defined as a specific subset of $\mathcal{D}$ intended for unlearning, denoted as $\mathcal{D}_{\text{fgt}}$. $\mathcal{D}_{\text{fgt}}$ contains copyrighted content targeted for removal or unlearning, i.e., $\mathcal{D}_{\text{fgt}} \subset \mathcal{D}$.

**Definition 3.2** (RETAIN SET).  A *retain set* is the remaining data, denoted as $\mathcal{D}_{\text{rt}}$, including all other data that is not targeted for removal and is crucial for maintaining the general knowledge and functionality of the model, i.e., $\mathcal{D}_{\text{rt}} = \mathcal{D} \setminus \mathcal{D}_{\text{fgt}}$.

**Gradient Ascent and Over-Forgetting**    Gradient ascent (GA) is a common process used for unlearning or removing the influence of the forget set $\mathcal{D}_{fgt}$ from an LLM. It aims to adjust the model's parameters $\theta$ to increase the loss associated with the $\mathcal{D}_{fgt}$, thereby diminishing the model's reliance on this data. Thus, its objective is to find a set of parameters that maximize the function,

$$\mathcal{L}_{fgt}(\theta) = -\mathbb{E}_{(x_i,y_i)\in\mathcal{D}_{\text{fgt}}}\left[\log \pi_{\theta_t}(y_i \mid x_i)\right], \tag{1}$$

where $(x_i, y_i)$ represents an input-output pair in $\mathcal{D}_{\text{fgt}}$. Then, GA updates the model's parameters $\theta$ by

$$\theta_{t+1} \leftarrow \theta_t + \eta \sum_{(x_i,y_i)\in\mathcal{D}_{\text{fgt}}} \nabla_{\theta_t}\mathcal{L}_{fgt}(\theta_t), \tag{2}$$

where $\eta$ is the learning rate for the GA loss.

Unlearning copyrighted content in an LLM often leads to the unintended consequence of *over-forgetting*—where not only the targeted copyrighted content is forgotten, but also related knowledge that is semantically similar yet does not infringe copyright law. This over-forgetting can significantly degrade the model's overall performance, especially as the need to unlearn more data arises in the future. More specifically, over-forgetting occurs when the loss maximization on $\mathcal{D}_{\text{fgt}}$ unintentionally affects nearby knowledge, increasing the loss in the embedding space close to $\mathcal{D}_{\text{fgt}}$. This problem happened due to the nature of GA, which diverges at a linear rate (Zhang et al., 2024), making it difficult to control the unlearning process precisely. Suppose that we have a set of knowledge $\mathcal{D}_x$ that is located between $\mathcal{D}_{\text{fgt}}$ and $\mathcal{D}_{\text{rt}}$, which represents the range also affected by GA during the unlearning of $\mathcal{D}_{\text{fgt}}$ (see the ring-shaped region in Figure 1). GA influences $\mathcal{D}_x$ through

$$\frac{\mathcal{L}_x(\theta)}{\mathcal{L}_{fgt}(\theta)} \propto \frac{c}{d(\mathcal{D}_x, \mathcal{D}_{\text{fgt}})}, \tag{3}$$

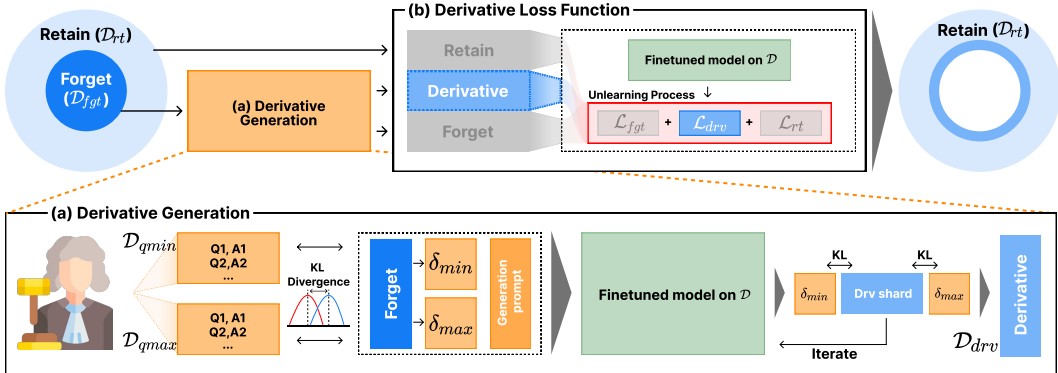

Figure 2: Overall procedure of MOUCHI. We first create the derivative set from the forget set through the **(a) derivative generation** module. Once the derivative set is obtained, MOUCHI performs unlearning with the **(b) derivative loss function** based on the generated derivative knowledge.

where $d(\mathcal{D}_x, \mathcal{D}_{\text{fgt}})$ represents the distance between $\mathcal{D}_x$ and $\mathcal{D}_{\text{fgt}}$ and $c$ is a constant. The closer $\mathcal{D}_x$ and $\mathcal{D}_{\text{fgt}}$ are, the more likely $\mathcal{D}_x$ is influenced by the unlearning process. In §4.2, we empirically verify this relationship in Eq. (3) .

# 4 PROPOSED LLM UNLEARNING FRAMEWORK: MOUCHI

## 4.1 PROBLEM STATEMENT

In this work, we aim to mitigate the problem of over-forgetting. First, we identify what constitutes $\mathcal{D}_x$, the set of data that are affected by over-forgetting. After identifying the affected set, the second step is to counteract over-forgetting by finding a set of parameter $\theta^*$ such that

$$\theta^* = \underset{\theta}{\operatorname{argmax}} \left[ \mathcal{L}_{fgt}(\theta) - \lambda \mathcal{L}_x(\theta) \right], \tag{4}$$

where $\lambda$ is a regularization parameter that controls the trade-off between the two losses. To solve the first step, we introduce the concept of derivative knowledge in §4.2. Furthermore, Figure 2 illustrates the overall framework of MOUCHI to mitigate over-forgetting. First, the framework begins by creating the derivative set from the forget set through derivative knowledge generation (§4.3). Once we have the derivative set, MOUCHI introduces a derivative loss function (§4.4) into the unlearning process. MOUCHI can be augmented on top of *any* existing unlearning methods.

## 4.2 DERIVATIVE KNOWLEDGE OVERVIEW

The *derivative knowledge*, denoted as $\mathcal{D}_{\text{drv}}$, refers to a set of knowledge that is *semantically similar* to the target knowledge to be removed in $\mathcal{D}_{\text{fgt}}$. While $\mathcal{D}_{\text{drv}}$ is closely related to $\mathcal{D}_{\text{fgt}}$, the content in $\mathcal{D}_{\text{drv}}$ is not considered copyright infringement and, therefore, must be preserved to maintain the overall performance of an LLM.

Let $\mathcal{P}_{\text{fgt}}$ and $\mathcal{P}_{\text{rt}}$ be the empirical distribution over the forget set $\mathcal{D}_{\text{fgt}}$ and the retain set $\mathcal{D}_{\text{rt}}$, respectively. Then, we have a derivative set $\mathcal{D}_{\text{drv}} = \{(x_i', y_i')\}$ sampled from the probability distribution $\mathcal{P}_{\mathcal{D}}$ of the dataset $\mathcal{D}$, where $(x_i', y_i')$ are distinct from any pairs $(x_i, y_i)$ in $\mathcal{D}_{\text{fgt}}$. Formally,

$$\mathcal{D}_{\text{drv}} = \{(x_i', y_i') \mid (x_i', y_i') \sim P_{\mathcal{D}}, \ (x_i', y_i') \notin \mathcal{D}_{\text{fgt}}\}, \tag{5}$$

where $0 \leq D_{KL}(\mathcal{D}_{\text{fgt}} \parallel \mathcal{D}_{\text{drv}}) \leq D_{KL}(\mathcal{D}_{\text{fgt}} \parallel \mathcal{D}_{\text{rt}})$. The KL divergence, $D_{KL}$, is used to measure the semantic similarity between two sets of knowledge. To ensure applicability in real-world copyright scenarios, we refine the equation by introducing arbitrary minimum and maximum bounds of

$$\delta_{\min} \leq D_{KL}(\mathcal{D}_{\text{fgt}} \parallel \mathcal{D}_{\text{drv}}) \leq \delta_{\max}. \tag{6}$$

These boundary values can be set by experts, such as lawmakers.

---

**Algorithm 1:** $\mathcal{D}_{\text{drv}}$ Generation Process

---

**Input:** QnA examples for $\delta_{\min}$ and $\delta_{\max}$, $\mathcal{D}_{\text{fgt}}$ (Forget Set), $\mathcal{D}_{\text{rt}}$ (Retain Set), prompt for
        generating additional QnAs (if needed)

**Output:** $\mathcal{D}_{\text{drv}}$ (derivative knowledge)

1 **Initialization:**
2     $D_{\text{qmin}}, D_{\text{qmax}} \leftarrow$ User provides example QnAs for the boundaries;
3     $\delta_{\min} \leftarrow D_{KL}(\mathcal{D}_{\text{fgt}} \parallel D_{\text{qmin}})$;
4     $\delta_{\max} \leftarrow D_{KL}(\mathcal{D}_{\text{fgt}} \parallel D_{\text{qmax}})$;
5 **while** $|\mathcal{D}_{drv}| < |\mathcal{D}_{fgt}|$ **do**
6     Generate a new QnA using the provided prompt;
7     Compute $D_{KL}(\mathcal{D}_{\text{fgt}} \parallel$ Generated QnA$)$;
8     **if** $\delta_{\min} \leq D_{KL}(\mathcal{D}_{fgt} \parallel \textit{Generated QnA}) \leq \delta_{\max}$ **then**
9        $\mathcal{D}_{\text{drv}} = \mathcal{D}_{\text{drv}} \cup$ Generated QnA;

10 **return** $\mathcal{D}_{\text{drv}}$;    # Will be used in the loss update for unlearning

---

**Empirical Evidence** In Figure 3, we examine the relationship between the KL distance of $\mathcal{D}_{\text{drv}}$ and $\mathcal{D}_{\text{fgt}}$ and their loss under vanilla GA to validate our hypothesis in Eq. (3). Each line in the graph represents the GA loss on its respective dataset, with the lines labeled "KL 0.1", "KL 0.2", "KL 0.3", and "KL 0.4" represent different KL divergence values between $\mathcal{D}_{\text{fgt}}$ and $\mathcal{D}_{\text{drv}}$. The results empirically confirm our hypothesis that the losses of the derivative sets increase as the distance to the forget set decreases, being more prone to accidental forgetting, as GA aims to find the maximum loss values. These results emphasize the importance of carefully managing the unlearning process to avoid over-forgetting.

## 4.3 DERIVATIVE KNOWLEDGE GENERATION

**Specifying the Boundary** The primary challenge in generating derivative knowledge is to first clearly define what should be included in $\mathcal{D}_{\text{drv}}$. For this purpose, we adopt a similar approach to that proposed by Vyas et al. (2023), which uses the KL divergence to quantify copyrighted content in generative models. Our method employs the KL divergence to measure the semantic closeness of the derivative set candidates to the forget data. However, we acknowledge that determining "fair use" falls outside our area of expertise. Instead, our objective is to programmatically generate derivative knowledge $\mathcal{D}_{\text{drv}}$ given a set of QnAs provided by experts, such as lawmakers.

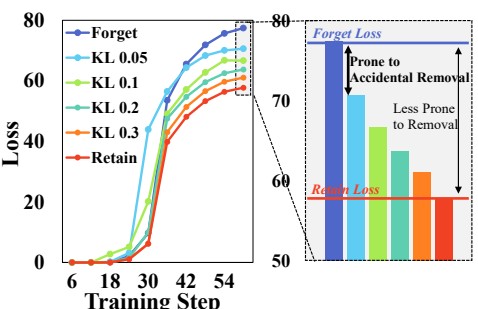

Figure 3: Under the vanilla GA setup, the loss on the $\mathcal{D}_{\text{drv}}$ with KL 0.05 grows significantly, second only to the $\mathcal{D}_{\text{fgt}}$. It indicates that vanilla GA negatively impacts derivative knowledge and leads to over-forgetting.

The process of defining the boundaries of derivative knowledge begins with the user, typically an expert, providing example QnAs that help establish the semantic boundaries. These boundaries are quantified using the KL divergence values. Specifically, the minimum boundary, $\delta_{\min}$, is determined by the KL divergence between the distribution of the forget data ($\mathcal{D}_{\text{fgt}}$) and a QnA that defines the lower boundary. Conversely, the maximum boundary, $\delta_{\max}$, is defined by the KL divergence between $\mathcal{D}_{\text{fgt}}$ and a QnA that establishes the upper boundary. That is,

$$\delta_{\min} = D_{KL}(\mathcal{D}_{\text{fgt}} \parallel \mathcal{D}_{\text{qmin}}) \quad \text{and} \quad \delta_{\max} = D_{KL}(\mathcal{D}_{\text{fgt}} \parallel \mathcal{D}_{\text{qmax}}). \tag{7}$$

**Obtaining Derivative Knowledge** Given a sufficiently comprehensive dataset $\mathcal{D}$, we can obtain $\mathcal{D}_{\text{drv}}$ defined in Eq. (7) as a subset of $\mathcal{D}_{\text{rt}}$. However, it is more likely that $\mathcal{D}_{\text{rt}}$ does not contain enough samples to generate $\mathcal{D}_{\text{drv}}$. Thus, we use $\delta_{\min}$ and $\delta_{\max}$ in conjunction with the forget set $\mathcal{D}_{\text{fgt}}$ to generate the derivative set $\mathcal{D}_{\text{drv}}$. For the generation model, we use the model fine-tuned on $\mathcal{D}$. The specific prompt used to generate $\mathcal{D}_{\text{drv}}$ is provided in §B.

As outlined in Algorithm 1, the process for generating $\mathcal{D}_{\text{drv}}$ is as follows.

1. **Incorporating the Forget Set and Boundary into the Prompt** (Line 6). The forget set $\mathcal{D}_{\text{fgt}}$ and the boundary values $\delta_{\min}$ and $\delta_{\max}$ are included in the prompt provided to the LLM for generating derivative knowledge $\mathcal{D}_{\text{drv}}$.

2. **Generating Derivative Knowledge** (Lines 5–9). The LLM generates a synthetic dataset to be used as derivative knowledge $\mathcal{D}_{\text{drv}}$. We keep generating until the size of $\mathcal{D}_{\text{drv}}$ is similar to $\mathcal{D}_{\text{fgt}}$

3. **Checking KL Divergence** (Lines 8–9). After creating $\mathcal{D}_{\text{drv}}$, we check the KL divergence of the generated shard to ensure that it is within the range defined by the boundary in Eq. (7).

This process facilitates the generation of derivative knowledge that maintains semantic relevance while adhering to fair use standards, as defined by the expert-provided QnAs.

## 4.4 DERIVATIVE LOSS FUNCTION

After obtaining sufficient data for $\mathcal{D}_{\text{drv}}$, we incorporate the generated $\mathcal{D}_{\text{drv}}$ at the beginning of the unlearning process. Its corresponding loss, $\mathcal{L}_{\text{drv}}$, is updated throughout the process. Hence, our complete unlearning loss function becomes

$$\theta_{t+1} \leftarrow \theta_t + w_{\text{fgt}}\nabla_{\theta_t}\mathcal{L}_{\text{fgt}}(\theta_t) - w_{\text{drv}}\nabla\mathcal{L}_{\text{drv}}(\theta_t) - w_{\text{rt}}\nabla\mathcal{L}_{\text{rt}}(\theta_t), \tag{8}$$

where each loss term, which represents the forget set, the derivative set, and the retain set, has a tunable weight denoted by $w_{\text{fgt}}$, $w_{\text{drv}}$, and $w_{\text{rt}}$, respectively. Note that the sign for the first term is $+$ for removal whereas the signs for the second and third terms are $-$ for retention. The specific loss functions used for each term are further discussed in §5.1.

## 5 EXPERIMENTS

### 5.1 EXPERIMENTAL SETTING

**Datasets**  We adopt **two** publicly available datasets, TOFU (Maini et al., 2024) and MUSE (Shi et al., 2024), to comprehensively validate the effectiveness of MOUCHI. Both datasets are specifically designed to facilitate unlearning in LLMs, making them particularly well-suited for our scenarios. The TOFU dataset consists of **200** fictional authors, with each author associated with **20** questions and answers about their books and personal information. The MUSE benchmark features two unlearning corpora: one derived from the Harry Potter series (MUSE-Books) and the other from BBC news articles (MUSE-News).

In our experiment, we use the full TOFU dataset and create a split where 25% of the data is designated as the forget set $\mathcal{D}_{\text{fgt}}$. We select a 25% split to increase the difficulty of the unlearning task as handling a larger forget set poses a more significant challenge. Also, smaller splits of 5% and 10% are explored in §5.4 to assess the impact of different forget set sizes on the unlearning process. For both the MUSE-News and MUSE-Books datasets, we use the entire forget and retain sets from the *knowmem* subset, following the original settings.

Due to the limited number of questions in the original datasets, we extend the dataset by generating the derivative set $\mathcal{D}_{\text{drv}}$ from the forget set $\mathcal{D}_{\text{fgt}}$, as described in §4.3. We generate an amount similar to the size of the forget set to ensure a balanced evaluation. We also modify the forget set to make the questions more closely resemble infringing content. Furthermore, for TOFU, we combine the *world_fact* and *real_author* subsets into a single normal set $\mathcal{D}_{nor}$ to assess the performance of the unlearned model on general knowledge questions since the results from existing approaches show no significant difference in performance between them.

**Comparison Baselines**  We explore different combinations of loss functions from the previous studies by having a linear combination of Eq. (8) as our baselines. For the **forget loss** ($\mathcal{L}_{\text{fgt}}$), we use all the losses in (Maini et al., 2024) and the NPO in Zhang et al. (2024). For the **retain loss** ($\mathcal{L}_{\text{rt}}$) and **derivative loss** ($\mathcal{L}_{\text{drv}}$), we use the vanilla GD and KL commonly used in prior studies (Wang et al., 2023; Chen & Yang, 2023). In addition to specific loss functions, we incorporate Task Vector (TV) proposed by Ilharco et al. (2023), which adjusts the model's behavior through simple arithmetic operations on its weights and has been employed in recent studies (Shi et al., 2024; Dou et al., 2024). The full explanations are provided in §C

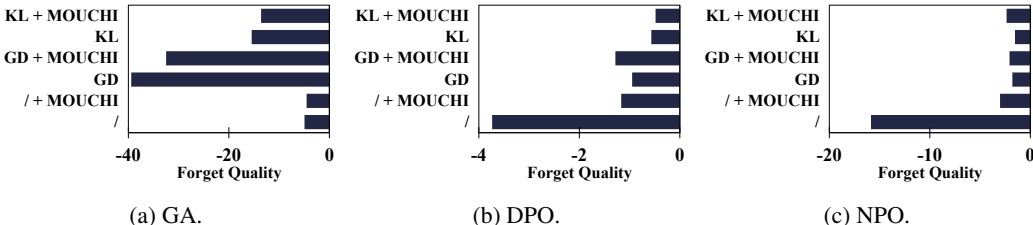

|          (a) GA.          |          (b) DPO.          |          (c) NPO.          |

Figure 4: Performance comparison between unlearning methods with and without our MOUCHI framework in terms of forget quality using KS-test's log $p$-value. Shorter bars indicate better results.

We include a wide range of state-of-the-art LLM unlearning methods for comparison. Maini et al. (2024) categorized the existing LLM unlearning methods *based on the loss functions used*. Following the categorization of Maini et al. (2024), we consider three forget losses $\mathcal{L}_{\text{fgt}}$ (GA, DPO, and NPO) with combinations that include a loss for $\mathcal{L}_{\text{rt}}$. Besides, we include an additional unlearning Task Vector (TV) framework. By default, $\mathcal{L}_{\text{drv}}$ introduced by MOUCHI is set to GD. However, if $\mathcal{L}_{\text{rt}}$ is used, $\mathcal{L}_{\text{drv}}$ will match it. For example, for a baseline with $\mathcal{L}_{\text{fgt}}$=NPO and $\mathcal{L}_{\text{rt}}$=KL, MOUCHI with $\mathcal{L}_{\text{drv}}$=KL is activated and compared against the baseline.

**Evaluation Metrics**  Evaluating the performance of unlearning methods is inherently challenging and remains an active area of research. To ensure that our evaluation is valid and comprehensible, we follow the guidelines outlined by Maini et al. (2024). We assess and compare the unlearned models using two key metrics: *model utility* and *forget quality*.

- **Model Utility** evaluates the model's ability to answer questions from each subset ($\mathcal{D}_{\text{fgt}}$, $\mathcal{D}_{\text{rt}}$, $\mathcal{D}_{\text{drv}}$, $\mathcal{D}_{\text{nor}}$). We use ROUGE (Lin, 2004), a widely adopted metric in natural language processing for sentence comparison. For the derivative, retain, and normal subsets, higher ROUGE scores indicate better performance, while for the forget subset, lower scores are better. However, it is important to note that a ROUGE score close to 0 on the forget subset suggests catastrophic collapse (Zhang et al., 2024), denoted as '*', meaning that the model produces nonsensical answers.
- **Forget Quality** is measured using the $p$-value from the Kolmogorov-Smirnov test (Maini et al., 2024). Intuitively, high $p$-values, where we cannot reject the null hypothesis that the two distributions are the same, indicate strong forgetting. Conversely, when the $p$-value is low, we can assert a difference between the unlearned model and the retained model, suggesting potential copyright infringement and poor unlearning. The main goal of unlearning is to obtain a model that replicates the performance of a model trained solely on $\mathcal{D}_{\text{rt}}$. To evaluate this aspect, we compare the outputs of the unlearned model with those of a model trained solely on $\mathcal{D}_{\text{rt}}$, using the truth values.

**Implementation Details**  The source code is available at https://anonymous.4open. science/r/MOUCHI. The experiment was conducted using the LLaMAv2-7B model (Touvron et al., 2023) with a batch size of 32 and a learning rate of $1 \times 10^{-4}$. All experiments were performed on two NVIDIA A6000 GPUs using optimization techniques, such as LoRA (Hu et al., 2022) and DeepSpeed (Rasley et al., 2020), to fit the model into the available GPU memory.

To specify $\delta_{\min}$ and $\delta_{\max}$ boundaries, we simulate expert input by using ChatGPT as a stand-in for a user, such as a lawmaker, who provides appropriate questions to determine $\delta_{\min}$ and $\delta_{\max}$. The prompt used for this simulation can be found in §B (see Figure 8).

## 5.2 MAIN RESULTS

**Mitigating Over-Forgetting in the Derivative Set**  Table 1 and Figure 4 present the model utility and forget quality for **20** different combinations on the TOFU dataset. Overall, the inclusion of MOUCHI plays a crucial role in preserving the derivative knowledge during the unlearning process across all baselines. Specifically, the model's utility score for the derivative set increases by **up to 145%** on the TOFU dataset when MOUCHI is used. For the MUSE benchmark (Table 2), MOUCHI achieves the model's utility score improvement of up to **94%**. Additionally, we observe that more stable forget loss algorithms lead to higher derivative utility scores. For example, the inclusion of the derivative loss achieves a score of 0.757 with the GA forget loss algorithm, while other configurations

Table 1: Model's utility score (ROUGE) comparison across multiple unlearning methods with and without our MOUCHI framework on the TOFU dataset. '*' indicates catastrophic collapse. The first column indicates the loss for $\mathcal{L}_{\text{fgt}}$, the second column that loss for $\mathcal{L}_{\text{rt}}$, the third column whether MOUCHI is activated with the corresponding configurations of $\mathcal{L}_{\text{fgt}}$ and $\mathcal{L}_{\text{rt}}$.

| $\mathcal{L}_{\text{fgt}}$ | $\mathcal{L}_{\text{rt}}$ | MOUCHI | Forget ↓ | Derivative ↑ | Retain ↑ | Normal ↑ |
|---|---|---|---|---|---|---|
| GA | none | ✗ | 0.001*±0.002 | 0.005*±0.004 | 0.007*±0.002 | 0.003*±0.001 |
| | | ✓ | **0.115±0.049** | **0.890±0.050** | **0.789±0.074** | **0.803±0.005** |
| | GD | ✗ | 0.005*±0.008 | 0.531±0.077 | 0.605±0.085 | **0.806±0.075** |
| | | ✓ | **0.057±0.034** | **0.757±0.079** | **0.733±0.082** | 0.744±0.054 |
| | KL | ✗ | 0.016*±0.011 | 0.158*±0.037 | 0.157*±0.036 | 0.049*±0.049 |
| | | ✓ | **0.122±0.077** | **0.890±0.060** | **0.169±0.089** | **0.772±0.065** |
| DPO | none | ✗ | 0.001*±0.001 | 0.004*±0.003 | 0.007*±0.002 | 0.003*±0.001 |
| | | ✓ | **0.241±0.107** | **0.992±0.019** | **0.729±0.001** | **0.799±0.003** |
| | GD | ✗ | **0.350±0.097** | 0.828±0.044 | **0.959±0.043** | 0.790±0.062 |
| | | ✓ | 0.496±0.109 | **0.981±0.025** | 0.786±0.121 | **0.801±0.108** |
| | KL | ✗ | 0.027*±0.038 | 0.813±0.055 | 0.921±0.066 | 0.777±0.078 |
| | | ✓ | **0.318±0.109** | **0.973±0.033** | **0.962±0.059** | **0.788±0.115** |
| NPO | none | ✗ | **0.391±0.039** | 0.393±0.072 | 0.383±0.076 | 0.598±0.024 |
| | | ✓ | 0.441±0.032 | **0.966±0.027** | **0.799±0.062** | **0.764±0.023** |
| | GD | ✗ | **0.514±0.029** | 0.854±0.088 | 0.868±0.072 | **0.778±0.034** |
| | | ✓ | 0.519±0.028 | **0.944±0.034** | **0.903±0.043** | 0.755±0.067 |
| | KL | ✗ | 0.581±0.030 | 0.576±0.043 | 0.675±0.068 | 0.751±0.051 |
| | | ✓ | **0.542±0.031** | **0.928±0.045** | **0.808±0.090** | **0.779±0.124** |
| TV | | ✗ | **0.762±0.011** | 0.752±0.071 | 0.735±0.054 | **0.491±0.031** |
| | | ✓ | 0.771±0.082 | **0.831±0.033** | **0.761±0.078** | 0.482±0.082 |
| Best Improvement | | | 6.7% | 145.8% | 108.6% | 28.1% |

Table 2: Model's utility score (ROUGE) comparison across multiple unlearning methods with and without our MOUCHI framework on the MUSE benchmark. '*' indicates catastrophic collapse. The first column indicates the loss for $\mathcal{L}_{\text{fgt}}$, the second column that loss for $\mathcal{L}_{\text{rt}}$, the third column whether MOUCHI is activated with the corresponding configurations of $\mathcal{L}_{\text{fgt}}$ and $\mathcal{L}_{\text{rt}}$.

| $\mathcal{L}_{\text{fgt}}$ | $\mathcal{L}_{\text{rt}}$ | MOUCHI | MUSE-Books Forget ↓ | Derivative ↑ | Retain ↑ | MUSE-News Forget ↓ | Derivative ↑ | Retain ↑ |
|---|---|---|---|---|---|---|---|---|
| GA | none | ✗ | 0.468±0.154 | 0.096*±0.075 | 0.034*±0.013 | 0.000*±0.000 | 0.000*±0.000 | 0.000*±0.000 |
| | | ✓ | **0.383±0.074** | **0.723±.199** | **0.844±0.107** | 0.000*±0.000 | **0.444±0.103** | **0.312±0.093** |
| | GD | ✗ | 0.648±0.133 | 0.913±0.034* | 0.924±0.066 | 0.043*±0.004 | 0.410±0.115 | 0.184±0.061 |
| | | ✓ | **0.542±0.056** | **0.970±0.012** | **0.924±0.031** | 0.047*±0.005 | **0.820±0.109** | **0.561±0.092** |
| | KL | ✗ | 0.623±0.089 | 0.952±0.021 | 0.942±0.070 | 0.000*±0.000 | 0.000*±0.000 | 0.000*±0.000 |
| | | ✓ | **0.601±0.045** | **0.996±0.006** | **0.988±0.014** | 0.080*±0.025 | **0.556±0.063** | **0.396±0.108** |
| DPO | none | ✗ | 0.012*±0.011 | 0.035*±0.034 | 0.332±0.112 | 0.002*±0.021 | 0.017*±0.107 | 0.001*±0.002 |
| | | ✓ | **0.196±0.087** | **0.990±0.017** | **0.429±0.118** | **0.261±0.057** | **0.970±0.031** | **0.782±0.079** |
| | GD | ✗ | 0.213±0.088 | 0.853±0.040 | **0.488±0.073** | **0.151±0.021** | 0.831±0.044 | 0.654±0.104 |
| | | ✓ | **0.182±0.038** | **0.970±0.072** | 0.225±0.092 | 0.192±0.039 | **0.978±0.043** | **0.765±0.081** |
| | KL | ✗ | 0.032*±0.128 | 0.902±0.052 | 0.905±0.064 | 0.010*±0.010 | 0.040*±0.035 | 0.021*±0.023 |
| | | ✓ | **0.511±0.081** | **0.991±0.010** | **0.938±0.026** | **0.278±0.060** | **0.733±0.066** | **0.639±0.057** |
| NPO | none | ✗ | 0.444±0.170 | 0.718±0.192 | 0.835±0.100 | 0.166±0.038 | 0.524±0.065 | 0.353±0.101 |
| | | ✓ | **0.375±0.153** | **0.993±0.008** | **0.981±0.023** | **0.138±0.027** | **0.881±0.042** | **0.576±0.086** |
| | GD | ✗ | **0.480±0.163** | 0.814±0.108 | 0.864±0.093 | **0.152±0.040** | 0.558±0.073 | 0.357±0.125 |
| | | ✓ | 0.496±0.159 | **0.993±0.008** | **0.947±0.046** | 0.214±0.029 | **0.893±0.091** | **0.869±0.113** |
| | KL | ✗ | 0.472±0.081 | 0.512±0.084 | 0.594±0.013 | 0.014*±0.010 | 0.061*±0.028 | 0.057*±0.036 |
| | | ✓ | **0.419±0.115** | **0.997±0.004** | **0.988±0.014** | **0.155±0.042** | **0.874±0.050** | **0.751±0.066** |
| TV | | ✗ | **0.831±0.072** | 0.852±0.033 | 0.811±0.041 | **0.741±0.025** | 0.619±0.042 | 0.736±0.102 |
| | | ✓ | 0.850±0.017 | **0.903±0.031** | **0.866±0.053** | 0.721±0.016 | **0.815±0.072** | **0.741±0.009** |
| Best Improvement | | | 14.5% | 94.7% | 66.3% | 28.9% | 60% | 143.4% |

surpass a score of 0.9. These results emphasize the importance of having better control over the unlearning process in mitigating over-forgetting.

Based on the main experimental results (Tables 1 and 2) from the TOFU and MUSE benchmarks, we further assess the effectiveness of the best-performing MOUCHI-augmented DPO on the MMLU benchmark to verify whether MOUCHI can maintain the general utility of the backbone LLaMA2-7B model, even after unlearning.

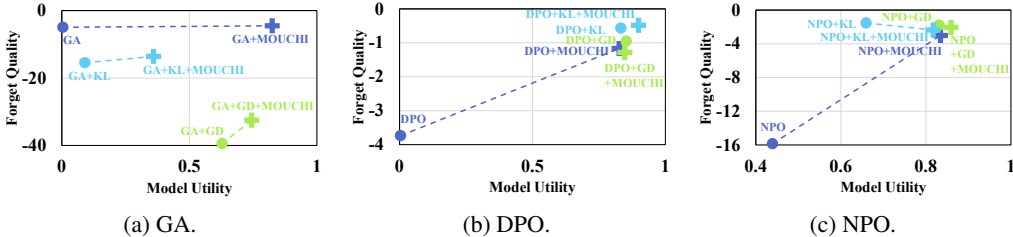

(a) GA.       (b) DPO.       (c) NPO.

Figure 5: Performance comparison of unlearning methods with and without MOUCHI in terms of model utility versus forget quality (log $p$-value). The upper right part is the better region.

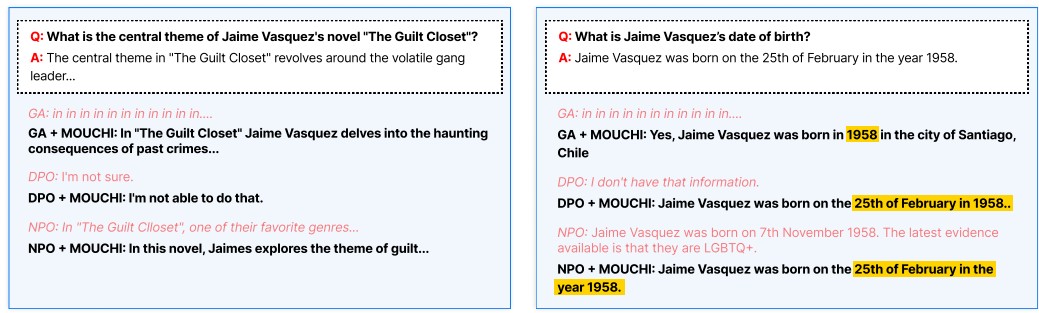

(a) Results on the Forget Set.       (b) Results on the Derivative Set.

Figure 6: Qualitative comparison on the forget and derivative sets between different methods.

As presented in Table 3, comparing to the reference utility score from LLaMA2-7B, we notice that the MMLU scores (Hendrycks et al., 2021) of the MOUCHI-augmented models are consistent across all datasets. These results indicate that MOUCHI effectively maintains general utility while mitigating the over-forgetting problem.

Table 3: Utility score comparison of MOUCHI-augmented models across different datasets on the MMLU benchmark.

| Dataset | STEM | Social Sciences | Humanities | Other | Average |
|---|---|---|---|---|---|
| TOFU | 0.372 | 0.510 | 0.424 | 0.511 | 0.452 |
| MUSE-Books | 0.380 | 0.549 | 0.431 | 0.543 | 0.472 |
| MUSE-News | 0.372 | 0.536 | 0.427 | 0.526 | 0.462 |
| Reference Score | 0.374 | 0.518 | 0.430 | 0.532 | 0.461 |

### 5.3 MORE IN-DEPTH ANALYSIS

**Better Control over the Unlearning Process**  MOUCHI provides better control over the unlearning process by slowing it down and preserving the derivative set. As shown in Table 1, catastrophic collapse occurs in the vanilla GA and DPO baselines. However, incorporating MOUCHI reduces the severity of catastrophic collapse in both cases while maintaining high utility across the remaining sets. Furthermore, this enhanced control improves the overall capability of the unlearned model. This improvement is evident in the utility scores of the normal and retain sets, where most of the performance of the derivative-augmented models surpasses its counterpart. Nevertheless, the inclusion of the derivative loss only mitigates the issue and does not eliminate it entirely. The forget set still achieves scores around 0.1 to 0.2, indicating that some outputs are still nonsensical.

**Better Model Utility and Forget Quality Trade-Off**  Figure 5 illustrates the trade-off between model utility and forget quality for all the unlearning methods. As observed from the figure, the methods augmented with MOUCHI generally perform better than their counterparts in terms of trade-off. Notably, despite the incorporation of $\mathcal{L}_{\mathrm{drv}}$, MOUCHI maintains a balance between model utility and forget quality. These results suggest that the inclusion of the derivative knowledge does not significantly degrade performance while mitigating the over-forgetting problem.

**Qualitative Results**  Figure 6a and 6b present example questions, expected answers, and generated answers from all baselines alongside their MOUCHI-augmented counterparts. Figure 6a shows that integrating MOUCHI with GA prevents catastrophic collapse. The results also show that MOUCHI closely mirrors the behavior of the underlying baseline models. For instance, DPO + MOUCHI generates the "I don't know" variant, similar to the original DPO model. Moreover, Figure 6b demonstrates that MOUCHI-augmented baselines consistently produce reasonable answers, successfully mitigating over-forgetting and maintaining model utility.

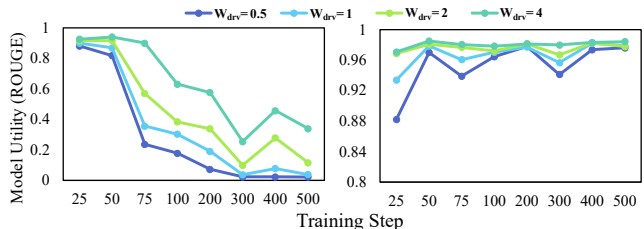

(a) Results on the Forget Set.    (b) Results on the Derivative Set.

Figure 7: Model utility over training steps of various $w_{\mathrm{drv}}$ values on the forget and derivative sets.

## 5.4 HYPERPARAMETER ANALYSIS

**Effect of $w_{\mathbf{drv}}$**    We observe the effect of varying the weight of the derivative knowledge loss on mitigating over-forgetting. Its value was chosen from 0.1, 0.5, 1, 2, and 4 to assess its impact. As shown in Figure 7, the inclusion of the derivative knowledge successfully mitigates over-forgetting across all tested values of $w_{\mathrm{drv}}$. Additionally, adding more weight to $\mathcal{L}_{\mathrm{drv}}$ further helps in preventing catastrophic collapse.

**Effect of KL Value for $\mathcal{L}_{\mathbf{drv}}$**    We conduct further experiments by varying the KL divergence between the derivative set and the forget set to examine its impact on both the forget set and the derivative set. Table 4 shows that when the KL divergence is small, such as 0.05 or 0.10, the model struggles to fully remove the influence of the forget set, as reflected in the model utility on the forget set and the forget quality. We conjecture that the derivative set, being semantically very close to the forget set, resembles it too closely, causing the loss updates from both sets to interfere with one another and partially cancel each other out. A potential solution to this issue is to assign more weight to the derivative loss, thereby placing greater emphasis on preserving the derivative set. For larger KL divergence values, however, the model returns to its usual performance.

Table 4: Performance comparison of derivative inclusion at different derivative set distances.

| KL Distance | Model Utility (ROUGE) | | | | Forget Quality (log $p$-value) |
|---|---|---|---|---|---|
| | Forget | Derivative | Retain | Normal | |
| 0.05 | 0.648 | 0.520 | 0.497 | 0.884 | -12.88 |
| 0.1 | 0.603 | 0.672 | 0.514 | 0.880 | -9.65 |
| 0.2 | 0.537 | 0.832 | 0.630 | 0.878 | -4.95 |
| 0.3 | 0.496 | 0.981 | 0.786 | 0.801 | -4.93 |

**Effect of Forget Split Size**    In the main experiment, we used a forget split of 25% as it presents a more challenging scenario for achieving strong results. Here, we also explore other forget splits of 5% and 10%. In general, MOUCHI works well on smaller splits. Similar to the 25% split's best improvement for the derivative set in Table 1, all smaller splits exhibit improvement of over 100%—115.2% for the 5% and 280% for the 10% split. The full results are provided in §D.

## 6 CONCLUSION

This paper proposes MOUCHI, a novel unlearning framework for LLMs, designed to mitigate the over-forgetting effects when unlearning copyrighted content. We introduce the concept of the *derivative knowledge*, which refers to a subset of information related to the target copyrighted content that must be preserved to adhere to fair use standards. By integrating the derivative knowledge into the existing unlearning algorithms via the derivative loss function, MOUCHI effectively reduces the over-forgetting problem, preserving valuable knowledge while enhancing the model's robustness across various tasks. Results from extensive experiments demonstrate that MOUCHI not only prevents the loss of related information but also improves the generalization capabilities of the model being unlearned, offering a balanced solution that respects copyright obligations while ensuring comprehensive and functional models.

## ETHICS STATEMENT

In accordance with the ICLR Code of Ethics, we are committed to the responsible stewardship of research, ensuring that our work adheres to the ethical principles outlined by the conference. Given the focus of our research on copyrighted content, we have taken special care to ensure that the dataset used in our experiments does not contain real copyrighted material, relying solely on publicly available synthetic datasets. Furthermore, our work is guided by the ethical imperative to respect the rights of copyright holders, particularly in the evolving landscape of LLMs, and aims to promote responsible and fair use of data.

## REPRODUCIBILITY STATEMENT

In accordance with ICLR's reproducibility guidelines, we provide an anonymized repository containing the code used in our experiments: https://anonymous.4open.science/r/MOUCHI.

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

# Appendix

## A    RELATED WORK ON GENERAL MACHINE UNLEARNING

**Exact Unlearning**   The concept of Machine Unlearning has emerged as a response to the need to remove specific knowledge from trained models without retraining them from scratch. Early attempts at machine unlearning sought to achieve exact unlearning, aiming to eliminate the influence of targeted data from the model completely. Papers such as Cao & Yang (2015), Ginart et al. (2019), and Brophy & Lowd (2021) tried to solve exact unlearning on specific algorithms such as Naive Bayes, k-means and random forest, respectively. Furthermore, Bourtoule et al. (2021) proposes a method to partition datasets into chunks to make the unlearning process more effective. However, these methods, as highlighted in various studies Xu et al. (2024); Thudi et al. (2022), have proven to be impractical. They are not only time-consuming and algorithm-specific but also require direct access to the initial training datasets—a requirement that is often unfeasible, especially in the context of LLMs, where datasets are vast and not always readily accessible.

**Approximate Unlearning**   Acknowledging these limitations, the focus has shifted towards approximate unlearning, a strategy aimed at reducing the influence of specific data on the model without the necessity for complete data removal. Papers such as (Golatkar et al., 2020; Guo et al., 2020; Marchant et al., 2022) utilize variations of the influence function, introduced by Koh & Liang (2017) to remove the influence of the forget data. Furthermore, more recent approaches (Thudi et al., 2022; Neel et al., 2021) use gradient ascent variations to make the unlearning process more efficient and algorithm-agnostic. Although promising, pure approximate unlearning seems unfeasible for use in LLM settings.

## B    PROMPTS FOR EXPERIMENTS

---

**Prompt use to generate Dqmin and Dqmax**

**Prompt:** I will employ the attached dataset in conjunction with another sets of Question and Answers delineated by two parameters: delta_min and delta_max, whose definitions are provided in the attached text.

Delta_min will encompass questions that are closely aligned with the author Q&A dataset while meticulously avoiding copyright infringement. Conversely, delta_max will define the boundary between knowledge derived from this dataset and entirely unrelated content.

Assume the role of a lawmaker capable of defining these boundaries and decide the suitable content to be included within the delta_min and delta_max limits. Your task is to generate x such content for both delta_min and delta_max

---

Figure 8: ChatGPT prompt to act as a lawmaker.

---

**Prompt use to generate Derivative Knowledge**

**Prompt:**  You are generating a dataset named 'Derivative Knowledge Dataset.' This dataset is derived from an existing dataset originally designed for unlearning copyrighted content related to book authors. The original dataset includes question and answer pairs about the authors and their book content. To effectively create the new dataset while adhering to fair use, include snippets from the original dataset as references.

Instructions:
  1. Inclusion of Original Data Snippets: Provide a snippet from the original dataset as a reference for each new question and answer pair you create. Ensure that these snippets are used to derive broader, non-specific questions that fall within legal bounds.

        example of unlearning data:
        {csv_files}

  2. Delta Bounds: Create questions based on two categories - delta_max (upper boundary of knowledge that can be retained without infringement) and delta_min (lower boundary of essential knowledge). Assume a KL divergence of X for delta_max and Y for delta_min between this new set and the original dataset, where X and Y are your specific KL divergence values.
  3. KL Divergence Use: The KL divergence values provided ({delta_max} for delta_max and {delta_min} for delta_min) guide the specificity and depth of your questions, ensuring they fall within the legal bounds of derivative knowledge.

---

Figure 9: Prompt used for derivative knowledge generation.

## C   LOSS FUNCTIONS

### C.1   FORGET LOSSES

For the **forget loss** ($\mathcal{L}_{\text{fgt}}$), we use the following optimization algorithms.

**GA** is the vanilla gradient ascent performed on $\mathcal{D}_{\text{fgt}}$. It is a widely used algorithm in the machine unlearning field due to its simplicity. GA essentially reverses the gradient descent (GD) process on $\mathcal{D}_{\text{fgt}}$. In our approach, we performed GA on $\mathcal{D}_{\text{fgt}}$ as

$$\mathcal{L}_{\text{GA}}(\theta_t) = \mathbb{E}_{(x_i, y_i) \in D_{\text{fgt}}} \left[ \log \pi_{\theta_t}(y_i \mid x_i) \right]. \tag{9}$$

**DPO** (Maini et al., 2024) is a modified version of direct preference optimization (Rafailov et al., 2023) adapted for unlearning. In this approach, "I don't know" or its variants are treated as positive responses, while answers in the forget set are considered negative responses. The corresponding loss function is deployed as

$$\mathcal{L}_{\text{DPO}}(\theta_t) = \mathbb{E}_{x_{\text{idk}} = [x, y_{\text{idk}}] \in D_{\text{fgt}}^{\text{idk}}} \left[ \log \pi_{\theta_t}(y_{\text{idk}} \mid x) \right]. \tag{10}$$

**NPO** Zhang et al. (2024) refers to negative preference optimization proposed to mitigate the problem of catastrophic collapsing. The loss also use DPO as their inspiration. We performed NPO on $\mathcal{D}_{\text{fgt}}$ as

$$\mathcal{L}_{\text{NPO},\beta}(\theta_t) = \frac{2}{\beta} \mathbb{E}_{(x_i, y_i) \in D_{\text{fgt}}} \left[ \log \left( 1 + \left( \frac{\pi_{\theta_t}(y_i \mid x_i)}{\pi_{\text{ref}}(y_i \mid x_i)} \right)^{\beta} \right) \right]. \tag{11}$$

### C.2   DERIVATIVE AND RETAIN LOSSES

For the **retain loss** ($\mathcal{L}_{\text{rt}}$) and **derivative loss** ($\mathcal{L}_{\text{drv}}$), we use the GD and KL algorithms.

**GD** is the vanilla gradient descent on $D_{\text{rt}}$, defined by

$$\mathcal{L}_{\text{GD}}(\theta_t) = \mathbb{E}_{(x_i, y_i) \in D_{\text{rt}}} \left[ \log \pi_{\theta_t}(y_i \mid x_i) \right]. \tag{12}$$

**KL** is commonly used in prior studies (Wang et al., 2023; Chen & Yang, 2023) to preserve the performance of $\mathcal{D}_{\text{rt}}$ by minimizing the difference between the original model and the model during unlearning. This is done by comparing the predictions on $\mathcal{D}_{\text{rt}}$ in the current unlearning process with those from the initial model (oracle model). The loss is defined by

$$\mathcal{L}_{\text{KL}}(\theta_t) = \frac{1}{|D_{\text{rt}}|} \sum_{(x_j, y_j) \in D_{\text{rt}}} D_{KL} \left( \pi_{\text{original}}(y_j \mid x_j) \parallel \pi_{\theta_t}(y_j \mid x_j) \right). \tag{13}$$

### C.3   TASK VECTOR

We adopted the approach from Ilharco et al. (2023). The method used straightforward arithmetic on the model weights, which can alter the behaviour of a model. Similar to Shi et al. (2024), We adapt task vectors to perform unlearning in two stages. To begin, $f_{\text{target}}$ is trained on $\mathcal{D}_{\text{fgt}}$ until the model is deliberately overfitted, resulting in a strengthened model $f_{\text{reinforce}}$. Next, a task vector associated with $\mathcal{D}_{\text{fgt}}$ is computed by finding the weight difference between $f_{\text{reinforce}}$ and $f_{\text{target}}$. Unlearning is then achieved by subtracting this task vector from $f_{\text{target}}$'s weights, redirecting the model away from the adaptation it had learned from $D_{\text{fgt}}$.

$$f_{\text{unlearn}} = f_{\text{target}} - (f_{\text{reinforce}} - f_{\text{target}}). \tag{14}$$

## D   FULL RESULT ON SMALLER FORGET SPLIT

We repeated the same experiment from Table 1 and Figure 4 using the smaller forget splits of 5% and 10%, as shown in Tables 5 and 6, respectively. Comparing these results with those in Table 1, it is shown that the effect of adding $\mathcal{L}_{\text{drv}}$ is significantly stronger in smaller splits, as evidenced by the greater ROUGE score improvement on the derivative set—115.2% and 280% for the 5% and 10% splits, respectively, compared to 145% for the larger split in Table 1.

Table 5: Performance comparison on the 5% forget split. '*' indicates catastrophic collapse.

| $\mathcal{L}_{\text{fgt}}$ | $\mathcal{L}_{\text{rt}}$ | MOUCHI | Forget ↓ | Derivative ↑ | Retain ↑ | Normal ↑ | Forget Quality |
|---|---|---|---|---|---|---|---|
| GA | none | ✗ | 0.097*±0.028 | 0.000*±0.000 | 0.000*±0.000 | 0.001*±0.004 | -19.17*±2.4 |
| | | ✓ | **0.081±0.019** | **0.748±0.115** | **0.645±0.116** | **0.288±0.213** | **-12.52±1.3** |
| | GD | ✗ | **0.465±0.044** | 0.490±0.101 | 0.656±0.089 | 0.739±0.141 | -4.52±0.7 |
| | | ✓ | 0.480±0.053 | **0.661±0.093** | **0.719±0.091** | **0.743±0.133** | **-3.73±0.6** |
| | KL | ✗ | 0.000*±0.000 | 0.000±0.000 | 0.000*±0.000 | 0.000*±0.000 | -54.2*±0.2 |
| | | ✓ | **0.127±0.032** | **0.824±0.084** | **0.398±0.041** | **0.514±0.092** | **-12.0±1.2** |
| DPO | none | ✗ | **0.251±0.079** | 0.610±0.087 | 0.607±0.082 | 0.811±0.097 | **-0.74±0.02** |
| | | ✓ | 0.342±0.065 | **0.967±0.034** | **0.891±0.087** | **0.812±0.097** | -1.05±0.09 |
| | GD | ✗ | 0.374±0.019 | 0.840±0.044 | 0.935±0.043 | **0.807±0.106** | -1.05±0.04 |
| | | ✓ | **0.326±0.018** | **0.990±0.011** | **0.948±0.040** | 0.805±0.105 | -1.05±0.1 |
| | KL | ✗ | 0.482±0.043 | 0.811±0.101 | 0.842±0.091 | **0.794±0.118** | **-0.84±0.09** |
| | | ✓ | **0.421±0.074** | **0.982±0.022** | **0.977±0.023** | 0.783±0.116 | -1.52±0.2 |
| NPO | none | ✗ | **0.450±0.049** | 0.414±0.083 | 0.421±0.082 | 0.687±0.207 | -11.8±1.3 |
| | | ✓ | 0.604±0.030 | **0.891±0.045** | **0.802±0.078** | **0.765±0.142** | **-4.32±0.8** |
| | GD | ✗ | **0.569±0.050** | 0.612±0.100 | 0.611±0.098 | 0.743±0.135 | -6.33±1.1 |
| | | ✓ | 0.590±0.041 | **0.910±0.054** | **0.905±0.054** | **0.779±0.120** | **-1.65±0.87** |
| | KL | ✗ | 0.460±0.051 | 0.506±0.104 | 0.508±0.099 | 0.710±0.157 | -15.07±2.3 |
| | | ✓ | **0.459±0.023** | **0.973±0.036** | **0.770±0.129** | **0.744±0.145** | **-3.01±0.5** |
| Best Improvement | | | 36.4% | 115.2% | 90.4% | 11.3% | 80% |

Table 6: Performance comparison on the 10% forget split. '*' indicates catastrophic collapse.

| $\mathcal{L}_{\text{fgt}}$ | $\mathcal{L}_{\text{rt}}$ | MOUCHI | Forget ↓ | Derivative ↑ | Retain ↑ | Normal ↑ | Forget Quality |
|---|---|---|---|---|---|---|---|
| GA | none | ✗ | 0.075*±0.014 | 0.040*±0.009 | 0.040*±0.009 | 0.012*±0.016 | -3.73*±0.9 |
| | | ✓ | 0.075*±0.014 | 0.040±0.009* | 0.042±0.016 | 0.012*±0.023 | -3.73*±0.7 |
| | GD | ✗ | 0.010*±0.010 | 0.382±0.086 | 0.381±0.091 | 0.412±0.201 | -4.95±0.3 |
| | | ✓ | 0.011*±0.011 | **0.456±0.083** | **0.421±0.089** | **0.421±0.046** | **-1.40±0.4** |
| | KL | ✗ | 0.000*±0.000 | 0.000*±0.000 | 0.000*±0.000 | 0.020*±0.024 | -67.3*±0.002 |
| | | ✓ | **0.113±0.054** | **0.851±0.043** | **0.231±0.082** | **0.691±0.029** | **-13.4±0.4** |
| DPO | none | ✗ | 0.059*±0.055 | 0.256±0.094 | 0.144±0.038 | 0.559±0.300 | -2.68±0.4 |
| | | ✓ | **0.605±0.165** | **0.974±0.024** | **0.874±0.030** | **0.798±0.122** | **-0.48±0.08** |
| | GD | ✗ | 0.329±0.064 | 0.873±0.030 | 0.968±0.022 | **0.791±0.117** | -0.94±0.05 |
| | | ✓ | **0.305±0.030** | **0.980±0.024** | **0.980±0.024** | 0.771±0.133 | **-0.84±0.1** |
| | KL | ✗ | 0.354±0.092 | 0.937±0.055 | 0.929±0.059 | **0.800±0.120** | **-0.48±0.04** |
| | | ✓ | **0.328±0.050** | **0.960±0.027** | **0.958±0.023** | 0.782±0.131 | -0.56±0.1 |
| NPO | none | ✗ | **0.399±0.063** | 0.628±0.079 | 0.631±0.075 | **0.805±0.144** | -17.05±2.2 |
| | | ✓ | 0.451±0.032 | **0.962±0.032** | **0.956±0.032** | 0.779±0.121 | **-11.5±0.7** |
| | GD | ✗ | 0.556±0.023 | 0.872±0.052 | 0.868±0.055 | **0.808±0.085** | -7.09±0.4 |
| | | ✓ | **0.502±0.027** | **0.962±0.031** | **0.957±0.024** | 0.806±0.103 | **-6.33±0.6** |
| | KL | ✗ | **0.573±0.016** | 0.782±0.062 | 0.779±0.061 | **0.796±0.112** | -15.8±3.2 |
| | | ✓ | 0.575±0.017 | **0.903±0.047** | **0.859±0.047** | 0.791±0.118 | **-9.61±0.8** |
| Best Improvement | | | 9.7% | 280% | 506% | 42.7% | 82% |

