# OpenReview forum: "MOUCHI: Mitigating Over-forgetting in Unlearning Copyrighted Information"
_ICLR.cc/2025/Conference — Submitted to ICLR 2025_

### Official Review · Reviewer_gJPq · 2024-10-29

**Soundness:** 2
**Presentation:** 3
**Contribution:** 3
**Rating:** 3
**Confidence:** 3

**Summary:**

Copyright concerns in large language models (LLMs) have become especially prominent due to the widespread use of these models, making them more vulnerable to misuse. Unlearning is a potential solution for removing such content. However, existing unlearning methods often suffer from over-forgetting, where the process unintentionally erases knowledge that is similar to copyrighted content falling under fair use and should be preserved. There is a trade-off between unlearning and over-forgetting.

This paper analyzes and identifies the over-forgetting problem in current LLM unlearning methods, then introduces the concept of derivative knowledge, and applies it to the MOUCHI framework by proposing the $L_{drv}$ loss. Experimental results show that this method achieves better performance than previous unlearning methods.

**Strengths:**

This paper investigates the issue of over-forgetting in unlearning methods. Its strengths include:

1. Analyzing and identifying the over-forgetting problem in current LLM unlearning methods.
2. Proposing the concept of derivative knowledge, which can be viewed as a buffer zone between the knowledge to be forgotten and the knowledge that needs to be retained.
3. Carefully designing a derivative generation module, with experimental results surpassing those of previous unlearning methods.

**Weaknesses:**

1. There are inconsistencies in the paper's wording. For example, in line 17, it states, "...the concept of derivative knowledge, a subset of information derived from copyrighted content that must be retained during unlearning." However, in line 76, it says, "the concept of derivative knowledge—a subset derived from the target copyrighted information that needs to be removed."

2. More experiments are needed to validate the effectiveness of the method. Currently, the paper is based on only one language model and a relatively small dataset (TOFU), where each author has only 20 QA questions. Such results may have certain limitations. There are many existing, more comprehensive unlearning datasets available, and broader testing is required to assess the method's generalizability.

[1] Muse: Machine unlearning six-way evaluation for language models. arXiv preprint arXiv:2407.06460, 2024.

[2] The wmdp benchmark: Measuring and reducing malicious use with unlearning. arXiv preprint arXiv:2403.03218, 2024.

[3] RWKU: Benchmarking Real-World Knowledge Unlearning for Large Language Models. arXiv preprint arXiv:2406.10890, 2024.

**Questions:**

1. Since the TOFU dataset is relatively small, this paper divides it into four subsets (by augmenting part of the data). Could you provide a detailed distribution of these subsets?

2. Regarding the evaluation metric "Derivative," if MOUCHI is not included, will the model not be trained on $D_{drv}$ ? Or is $D_{drv}$ integrated into other subsets for training?

3. Given the diversity of ChatGPT's generated content, has the determination of δmin and δmax undergone multiple validations to ensure its accuracy?

4. In the example on the right side of Figure 6, is the given "A" the knowledge that the model needs to retain? If so, what specifically needs to be forgotten? For clarity, it would be helpful to include related Q&A from both the Forget set and the Derivative set for the same author in the presentation, illustrating what should be retained versus what should be forgotten.

---

> ### Author Response · Authors · 2024-11-20
> **Response to Reviewer gJPq (1/2)**
>
> We thank the reviewer for their insightful comments and the time spent reviewing our paper.
>
> > There are inconsistencies in the paper's wording. For example, in line 17, it states, "...the concept of derivative knowledge, a subset of information derived from copyrighted content that must be retained during unlearning." However, in line 76, it says, "the concept of derivative knowledge—a subset derived from the target copyrighted information that needs to be removed."
>
> **R1.** We apologize for the inconsistencies in the paper's wording. We have revised the derivative knowledge definition in line 76 to ensure the consistency.
>
> > More experiments are needed to validate the effectiveness of the method. Currently, the paper is based on only one language model and a relatively small dataset (TOFU), where each author has only 20 QA questions. Such results may have certain limitations. There are many existing, more comprehensive unlearning datasets available, and broader testing is required to assess the method's generalizability.
>
> **R2.** Thank you for your valuable feedback and for highlighting the importance for broader validation. We selected the TOFU dataset because it most closely aligns with our goal of unlearning copyrighted information, as no dataset specifically designed for this purpose currently exists. In order to make TOFU applicable to our problem setting, we modified the forget set to better reflect copyrighted content. To address your concerns, we added the MUSE dataset to our experiment because it is the only dataset that we can use to replicate copyrighted content unlearning setting, along with one additional baseline as also suggested by Reviewer [c6sN](https://openreview.net/forum?id=N2wPtFVK6o&noteId=k2029RKIGj). Please refer to Tables A1 and A2 in the global response for detailed results and analysis for the MUSE dataset, as well as Tables B1, B2, and B3 for the results of one additional baseline.
>
> For further clarifications regarding the other datasets you mentioned, we outline the inapplicability of these datasets in our problem setting below.
> 1. **WMDP Benchmark**: This dataset is tailored for unlearning hazardous information, such as content related to bioweapons and hacking. Therefore, it does not align with our focus on copyright information removal.
> 2. **RWKU**: This dataset aims at general unlearning tasks involving the removal of knowledge about authors through fill-in-the-blank questions. The content predominantly features general knowledge, which falls under fair use and does not adequately represent copyright-sensitive data.
>
> > Since the TOFU dataset is relatively small, this paper divides it into four subsets (by augmenting part of the data). Could you provide a detailed distribution of these subsets?
>
> **R3.** We use the full TOFU dataset and maintain the original distribution of the forget and retain sets. However, due to the limited availability of data and the relatively similar performance observed in existing studies, we combined the world_fact and real_authors subsets into a single set, denoted as the normal set (\\( \mathcal{D}\_{nor} \\)). For the derivative set (\\( \mathcal{D}\_{drv} \\) ), we generated an amount similar to the size of the forget set to ensure balanced evaluation.

---

> ### Author Response · Authors · 2024-11-20
> **Response to Reviewer gJPq (2/2)**
>
> > Regarding the evaluation metric "Derivative," if MOUCHI is not included, will the model not be trained on $D_{drv}? Or $D_{drv}$ is integrated into other subsets for training?
>
> **R4.** Thank you for your question. By default, the fine-tuned model incorporates the derivative set (\\( \mathcal{D}_{drv} \\)) as part of the retain set during training. For the evaluation phase, if MOUCHI is not applied, it indicates that the derivative loss is not activated during the unlearning process. We have included this clarification in the updated version of the paper.
>
> > Given the diversity of ChatGPT's generated content, has the determination of δmin and δmax undergone multiple validations to ensure its accuracy?
>
> **R5.** Thank you for your insightful question. To ensure accuracy in determining δmin and δmax, we conducted **multiple iterations** of boundary generation and calculated the **average values** from these runs. This approach helps mitigate variability and enhances reliability.
>
> > In the example on the right side of Figure 6, is the given "A" the knowledge that the model needs to retain? If so, what specifically needs to be forgotten? For clarity, it would be helpful to include related Q&A from both the Forget set and the Derivative set for the same author in the presentation, illustrating what should be retained versus what should be forgotten.
>
> **R6.** Thank you for your suggestion. We updated the figure using the same authors. While the actual derivative knowledge depends on the boundary values (\\( \delta_{min} \\) and \\( \delta_{max} \\)). In general, the forget set encompasses information regarding the author’s book. In contrast, the derivative knowledge typically comprises related content beyond the book, such as the author's profile and background information.

---

> ### Author Response · Authors · 2024-11-25
> **Gentle Reminder**
>
> Dear Reviewer gJPq,
>
> Thank you for your time and effort in reviewing our paper! Since the discussion period will conclude in a few days, we would greatly appreciate it if you could review our responses and share any additional feedback or thoughts before the deadline (on November 26th AoE).
>
> Best regards,
>
> Submission 7011 Authors

---

> ### Author Response · Authors · 2024-12-02
> **Follow-up on Reviewer Feedback**
>
> Dear Reviewer gJPq,
>
> Thank you for your time and thoughtful feedback. As we approach the end of the discussion period (approximately one day remaining), we noticed that you have not yet responded to our answers to your concerns.
>
> We have carefully addressed all your comments and made every effort to improve the quality of the work based on your valuable suggestions. These efforts include improving the paper's presentation by revising the problematic parts you highlighted, clarifying the details of the experimental setup, and conducting additional experiments on the suggested benchmark.
>
> Given these responses, we kindly ask if you would be willing to review our rebuttal and reconsider your rating. If there are any remaining concerns that need to be addressed to justify a higher score, we would greatly appreciate your feedback before the discussion period concludes.
>
> Thank you once again for your time and consideration.
>
> Best regards,
>
> Submission 7011 Authors

---

### Official Review · Reviewer_w5RU · 2024-11-01

**Soundness:** 3
**Presentation:** 3
**Contribution:** 2
**Rating:** 5
**Confidence:** 4

**Summary:**

To address the over-forgetting in LLM unlearning, this paper proposes MOUCHI, a new unlearning framework that generates a set of derivative knowledge to enrich the retain set. The derivative knowledge lies between the forget set and the retain set, which helps to better specify the boundaries of the knowledge to be removed. Experimental results demonstrate that MOUCHI can provide better control over over-forgetting.

**Strengths:**

1.This paper is well motivated as it focuses on the over-forgetting issue in unlearning.

2.This paper proposes a simple method to expand the retain set in the form of data synthesis.

**Weaknesses:**

1.The proposed method lacks novelty as it simply prompts the model to generate derivative knowledge and uses KL divergence for filtering. And there is no improvement in the loss function for the derivative set, which can be regarded as an expansion of the retain set. Is there a generation-free way to induce derivative knowledge?

2.This paper only conducts experiments on TOFU and needs to verify the effectiveness on more unlearning datasets, such as scenarios where the range of unlearning knowledge is particularly extensive.

3.Some implementation details are lacking. For example, the forget set in TOFU consists of 200 fictional authors and has no connection with real-world knowledge. Then what is the derivative knowledge of these fictional authors? Some generated results need to be provided.

**Questions:**

Please see the weakness.

---

> ### Author Response · Authors · 2024-11-20
> **Response to Reviewer w5RU**
>
> Thank you for your feedback. We have addressed each point raised in your review below.
>
> > The proposed method lacks novelty as it simply prompts the model to generate derivative knowledge and uses KL divergence for filtering. And there is no improvement in the loss function for the derivative set, which can be regarded as an expansion of the retain set.
>
> **R1.** We appreciate your concerns regarding our novelty. We would like to clarify that **our main contributions focus on identifying and addressing the over-forgetting problem in LLM unlearning, which, to the best of our knowledge, has not been explored in prior work**. This issue is critical for improving the reliability of copyrighted content removal.
>
> While we use KL divergence for filtering and retain the existing loss function, **the novelty of our approach is in introducing the concept of derivative knowledge to effectively mitigating over-forgetting issues**.
>
>
> > Is there a generation-free way to induce derivative knowledge?
>
> **R2.** As discussed in line 264, given a sufficiently comprehensive dataset, we can obtain derivative knowledge as a subset of the retain set. By sufficient, we mean a dataset that includes both infringing and non-infringing types of questions. However, at present, there is no publicly available dataset specifically designed for copyright removal, which limits non-iterative approaches. Consequently, we employ an iterative method to effectively address this challenge.
>
> > This paper only conducts experiments on TOFU and needs to verify the effectiveness on more unlearning datasets, such as scenarios where the range of unlearning knowledge is particularly extensive.
>
> **R3.** Thank you for your valuable feedback. To address your suggestion and those from other reviewers, we conducted additional experiments using more datasets, specifically MUSE-Books and MUSE-News, to further validate the performance of MOUCHI. Please refer to Tables A1 and A2 in the global response for detailed results and analysis, as well as Tables B1, B2, and B3 for the results of one additional baseline.
>
> > Some implementation details are lacking. For example, the forget set in TOFU consists of 200 fictional authors and has no connection with real-world knowledge. Then what is the derivative knowledge of these fictional authors? Some generated results need to be provided.
>
> **R4.** We have presented the implementation details of the model and datasets used in Section 5.1, including specifics of the forget set. While the actual derivative knowledge depends on the boundary values (\\( \delta_{min} \\) and \\( \delta_{max} \\)). In general, the forget set encompasses information regarding the author’s book. In contrast, the derivative knowledge typically comprises related content beyond the book, such as the author's profile and background information. Examples of forget and derivative sets are illustrated in Figure 6 in our paper.

---

> ### Author Response · Authors · 2024-11-25
> **Gentle Reminder**
>
> Dear Reviewer w5RU,
>
> Thank you for your time and effort in reviewing our paper! Since the discussion period will conclude in a few days, we would greatly appreciate it if you could review our responses and share any additional feedback or thoughts before the deadline (on November 26th AoE).
>
> Best regards,
>
> Submission 7011 Authors

---

> > ### Comment · Reviewer_w5RU · 2024-11-28
> >
> > Thanks for your responses. Based on the author's responses and the comments of other reviewers, I have decided to maintain the rating score.

---

> > > ### Author Response · Authors · 2024-12-02
> > > **Thank you for your feedback**
> > >
> > > Dear Reviewer w5RU,
> > >
> > > Thank you for your thoughtful engagement with our manuscript and for considering our rebuttal. We greatly appreciate your feedback and the opportunity to refine our work.
> > >
> > > We noticed that you decided to maintain your score. To help us address any remaining concerns more effectively, could you kindly clarify where the rebuttal and general response fall short of addressing your feedback? Additionally, we would be grateful for any specific suggestions on how we might bridge this gap. Your guidance would be invaluable in strengthening these areas and better aligning the work with your expectations.
> > >
> > > Thank you again for your time and constructive feedback. We look forward to hearing your insights during the discussion period.
> > >
> > > Best regards,
> > >
> > > Submission 7011 Authors

---

### Official Review · Reviewer_c6sN · 2024-11-03

**Soundness:** 2
**Presentation:** 3
**Contribution:** 2
**Rating:** 5
**Confidence:** 3

**Summary:**

The paper titled "MOUCHI: Mitigating Over-Forgetting in Unlearning Copyrighted Information" addresses the critical issue of copyright infringement within large language models (LLMs). Due to the extensive and often indiscriminate data used in their training, LLMs can inadvertently memorize and reproduce copyrighted content, posing a significant legal and ethical challenge. The proposed MOUCHI framework introduces "derivative knowledge" as a subset of information derived from copyrighted content but intended for retention during unlearning to maintain fair-use knowledge. MOUCHI integrates a derivative loss function to differentiate and preserve derivative knowledge, allowing the model to continue answering general questions related to copyrighted material while removing infringing content. The framework is designed to work seamlessly with existing unlearning methods, providing a plug-and-play solution. Experimental results show that MOUCHI successfully reduces unintended knowledge loss by up to 145% over baseline methods.

**Strengths:**

The paper is well-written, with a clear explanation of the problem and the proposed solution.

The concept of over-forgetting and derivative knowledge is effectively introduced and explained, making the paper accessible to a broad audience.

Copyright concerns in LLMs are increasingly pressing, particularly as these models become more widely deployed. The focus on balancing copyright compliance with preserving relevant knowledge is timely and important. MOUCHI attempts to address the complex need to avoid copyright infringement while retaining content that may fall under fair use.

**Weaknesses:**

The MOUCHI framework, while straightforward, maybe too simplistic to fully address the nuanced requirements of copyright compliance.

MOUCHI may still risk indirect copyright violations by retaining derivative knowledge that could closely resemble the copyrighted content "CopyBench: Measuring Literal and Non-Literal Reproduction of Copyright-Protected Text in Language Model Generation"

Utility evaluation is limited, which weakens the claim of solving overforgetting problems. The authors may need to run common benchmarks such as MMLU to ensure the utility of LLM is not sabotaged.

The paper does not compare MOUCHI with some recent and relevant unlearning baselines, such as Goldfish Loss "Be like a goldfish, don’t memorize!" and "Avoiding Copyright Infringement via Machine Unlearning"

Code is not given.

Although the authors reference several key studies, the paper could be strengthened by discussing additional recent works that are pertinent to the topic of copyright compliance in LLMs. Listed below:

- Foundation Models and Fair Use
- Rethinking LLM Memorization through the Lens of Adversarial Compression
- Llms and memorization: On quality and specificity of copyright compliance
- SHIELD: Evaluation and Defense Strategies for Copyright Compliance in LLM Text Generation
- Digger: Detecting copyright content misusage in large language model training
- Speak, Memory: An Archaeology of Books Known to ChatGPT/GPT-4

**Questions:**

KL Divergence value is constrained by ChatGPT? Is this possible without an iterative approach?

How to know whether the generated dataset is correct and does not include hallucinations?

---

> ### Author Response · Authors · 2024-11-20
> **Response to Reviewer c6sN (1/2)**
>
> We thank the reviewer for their thorough comments and questions.
>
> > The MOUCHI framework, while straightforward, maybe too simplistic to fully address the nuanced requirements of copyright compliance.
>
> **R1.** We acknowledge your concern regarding the perceived simplicity of the MOUCHI framework in addressing the complexities of copyright compliance. We also recognize that copyright compliance is multifaceted, involving various legal, ethical, and operational considerations. Our intention was to create a novel, model-agnostic framework that can be seamlessly integrated into existing unlearning methods. Its plug-and-play design ensures compatibility with diverse unlearning scenarios, from targeted removal of specific contents to compliance with broader regulatory standards like the EU AI Act. That is, **the MOUCHI framework is designed to be flexible, allowing for the integration of additional layers of complexity as techniques for quantifying potential copyright infringement evolve**.
>
>
> > MOUCHI may still risk indirect copyright violations by retaining derivative knowledge that could closely resemble the copyrighted content "CopyBench: Measuring Literal and Non-Literal Reproduction of Copyright-Protected Text in Language Model Generation"
>
> **R2.** Thank you for your concern regarding the risk of indirect copyright violations. Our framework, MOUCHI, is specifically designed to address this issue. Derivative knowledge is defined using carefully controlled semantic boundaries *in the embedding space*, as quantified by KL divergence metrics. These boundaries (\\( \delta_{min} \\), \\( \delta_{max} \\) ) are set to ensure that the derivative knowledge is **semantically distinct** from the copyrighted content.
>
>
> > Utility evaluation is limited, which weakens the claim of solving overforgetting problems. The authors may need to run common benchmarks such as MMLU to ensure the utility of LLM is not sabotaged.
>
> **R3.** Thank you for your valuable feedback regarding utility evaluation. In our work, the TOFU dataset includes the *world_fact* and *real_authors* subsets, which we combined into the normal set (\\( \mathcal{D}_{nor} \\)) to evaluate the general utility of the LLM, similar to the intention of the MMLU benchmark. In response to your suggestion, we conducted additional experiments using the MMLU benchmark for the best-performing MOUCHI-augmented model, DPO, (selected based on the performance on TOFU and MUSE benchmarks) across all datasets. The results are as follows.
>
> | **Dataset**   | **STEM** | **Social Sciences** | **Humanities** | **Other** | **Average** |
> |:---------------|:----------:|:---------------------:|:----------------:|:-----------:|:-------------:|
> | TOFU          | 0.372    | 0.510                | 0.424          | 0.511     | 0.452       |
> | MUSE-Books    | 0.380     | 0.549               | 0.431          | 0.543     | 0.472       |
> | MUSE-News     | 0.372    | 0.536               | 0.427          | 0.526     | 0.462       |
> | **Reference Score (LLaMA2-7B)**  | 0.374    | 0.518               | 0.430           | 0.532     | 0.461       |
>
> These results indicate that MOUCHI effectively maintains general utility (after unlearning) while mitigating the over-forgetting problem.

---

> ### Author Response · Authors · 2024-11-20
> **Response to Reviewer c6sN (2/2)**
>
> > The paper does not compare MOUCHI with some recent and relevant unlearning baselines, such as Goldfish Loss "Be like a goldfish, don’t memorize!" and "Avoiding Copyright Infringement via Machine Unlearning"
>
> **R4.** Thank you for pointing out the recent papers we missed. We have made considerable efforts to compare MOUCHI with recent baselines, including NPO and KL-based methods. In response to your feedback and the baseline from a paper [1] suggested by Reviewer [gJPq](https://openreview.net/forum?id=N2wPtFVK6o&noteId=IjJoHmgQ8q), we have included an additional baseline—Task Vector [1], as used in the *Avoiding Copyright Infringement via Machine Unlearning* paper [2]. This baseline has been integrated into our evaluations on the existing datasets, as well as a newly added dataset. The results are as follows.
>
> **TV on TOFU**
> | MOUCHI | Forget ↓     | Derivative ↑  | Retain ↑     | Normal ↑     |
> |--------|--------------|---------------|--------------|--------------|
> | ✗      | **0.762±0.011**  | 0.752±0.071   | 0.735±0.054  | **0.491±0.031**  |
> | ✓      | 0.771±0.082  | **0.831±0.033**   | **0.761±0.078** | 0.482±0.082  |
>
> **TV on MUSE-Books**
> | MOUCHI | Forget ↓     | Derivative ↑  | Retain ↑     |
> |--------|--------------|---------------|--------------|
> | ✗      | **0.831±0.072**  | 0.852±0.033   | 0.811±0.041  |
> | ✓      | 0.850±0.017  | **0.903±0.031**   | **0.866±0.053**  |
>
> **TV on MUSE-News**
> | MOUCHI | Forget ↓     | Derivative ↑  | Retain ↑     |
> |--------|--------------|---------------|--------------|
> | ✗      | 0.741±0.025  | 0.619±0.042   | 0.736±0.102  |
> | ✓      | **0.721±0.016**  | **0.815±0.072**   | **0.741±0.009**  |
>
> However, we did not include the second paper, *Be like a goldfish, don’t memorize!*, as a baseline. This approach is designed to prevent next-token prediction from generating verbatim copies of stored knowledge, which makes it more suitable for addressing general memorization issues rather than the specific challenge of copyright information removal. Therefore, we believe it is beyond the scope of our paper.
>
> > Code is not given.
>
> **R5.** We published our code anonymously at https://anonymous.4open.science/r/MOUCHI. The link is also included in the Reproducibility Statement section of our paper.
>
> > Although the authors reference several key studies, the paper could be strengthened by discussing additional recent works that are pertinent to the topic of copyright compliance in LLMs.
>
> **R6.** We appreciate the reviewer’s suggestion to incorporate additional recent works on copyright compliance in LLMs. In response, we have expanded the Related Work section to include and discuss these studies. Thank you for pointing this out.
>
>
> > KL Divergence value is constrained by ChatGPT? Is this possible without an iterative approach?
>
> **R7.** As discussed in line 264 of the revised paper, given a sufficiently comprehensive dataset, we can obtain derivative knowledge as a subset of the retain set. By sufficient, we mean a dataset that includes both infringing and non-infringing types of questions. However, at present, there is no publicly available dataset specifically designed for copyright removal, which limits non-iterative approaches. Consequently, we employ an iterative method to effectively address this challenge.
>
> > How to know whether the generated dataset is correct and does not include hallucinations?
>
> **R8.** Thank you for your thoughtful question. We acknowledge the reviewer’s concern regarding potential hallucinations in the generated dataset. However, to mitigate this risk, we use a *fine-tuned* LLaMA2-7B model for each specific dataset, which helps align the outputs closely with the underlying data distribution. **Additionally, please note that hallucinated outputs typically exhibit a significant deviation from the original forget set, resulting in higher KL divergence values.** This type of output is systematically identified and filtered through our iterative boundary-checking process.
>
> **References**
>
> [1] Gabriel Ilharco *et al*. Editing models with task arithmetic. In *Proceedings of the 11th International Conference on Learning Representations (ICLR)*, 2023
>
> [2] Guangyao Dou *et al*. Avoiding copyright infringement via machine unlearning. *arXiv preprint arXiv:2406.10952*, 2024.

---

> ### Author Response · Authors · 2024-11-25
> **Gentle Reminder**
>
> Dear Reviewer c6sN,
>
> Thank you for your time and effort in reviewing our paper! Since the discussion period will conclude in a few days, we would greatly appreciate it if you could review our responses and share any additional feedback or thoughts before the deadline (on November 26th AoE).
>
> Best regards,
>
> Submission 7011 Authors

---

> ### Comment · Reviewer_c6sN · 2024-12-02
>
> Thank you for your responses. I have reviewed the author's response and the comments from other reviewers and have decided to maintain my original rating.

---

> > ### Author Response · Authors · 2024-12-02
> > **Follow-Up Response to Reviewer c6sN**
> >
> > Dear Reviewer c6sN,
> >
> > We sincerely appreciate your thoughtful evaluation of our work and understand your decision to maintain your original rating. To ensure we address any remaining concerns comprehensively, could you kindly clarify specifically where our rebuttal and general response fell short in addressing your concerns? Additionally, we would greatly value your guidance on how we can bridge this gap.
> >
> > If there are particular issues or areas for improvement that you believe are critical to revising your assessment, we are eager to hear them and incorporate your insights. We remain fully committed to clarifying any uncertainties and providing additional evidence to strengthen the paper before the discussion period concludes.
> >
> > Thank you once again for your time and constructive feedback.
> >
> > Best regards,
> >
> > Submission 7011 Authors

---

### Official Review · Reviewer_bVJx · 2024-11-04

**Soundness:** 3
**Presentation:** 3
**Contribution:** 3
**Rating:** 6
**Confidence:** 3

**Summary:**

This paper tackles the problem of over-forgetting during the unlearning process of LLMs, i.e., unintentionally removing similar content which should be preserved. The authors propose a concept of derivative knowledge, which is a subset of information derived from the copyrighted content and should be retained during unlearning. In particular, 1) the set derivative knowledge will be generated and 2) a derivative loss function is included. Different unlearning approaches can incorporate their proposed method and obtain various performance improvements in terms of model utility and forget quality.

**Strengths:**

* Propose an interesting and effective approach to tackle the over-forgetting issue of LLMs' unlearning process
* The approach can be integrated into existing unlearning methods to achieve various performance improvements
* Detailed analysis of the experiment results

**Weaknesses:**

* Lack of manual analysis of the derivative knowledge: the authors also mention that experts/lawmakers need to be involved to check the boundary, otherwise it is difficult to judge how good (enough) the KL-based semantic similarity is in this context. I would suggest the authors to incorporate human judgements in this process.
* Experiments are restricted to one dataset, also one scenario/domain. Maybe the authors could comment on how easy/difficult for the approach to be adapted to other types of copyrighted information.

**Questions:**

See weaknesses.

---

> ### Author Response · Authors · 2024-11-20
> **Response to Reviewer bVJx**
>
> Thank you for your valuable and positive feedback. We provide the following response to address your concerns.
>
> > Lack of manual analysis of the derivative knowledge: the authors also mention that experts/lawmakers need to be involved to check the boundary, otherwise it is difficult to judge how good (enough) the KL-based semantic similarity is in this context. I would suggest the authors to incorporate human judgements in this process.
>
> **R1.** We understand and appreciate the importance of incorporating human judgment in evaluating the boundary of derivative knowledge. In this work, we tried our best to approximate human judgment by leveraging the latest version of ChatGPT and conducted repeated experiments to ensure consistency in performance. While we acknowledge that relying solely on ChatGPT has its limitations, it offers reproducibility and scalability that would not be feasible if the evaluation relied on human judgment alone.
>
>
> > Experiments are restricted to one dataset, also one scenario/domain. Maybe the authors could comment on how easy/difficult for the approach to be adapted to other types of copyrighted information.
>
> **R2.** Currently, there is no dataset specifically designed for the task of removing copyrighted information, as most available datasets cater to general unlearning experiments. We believe that an ideal dataset for this purpose would include both infringing and non-infringing samples, which can be similar in nature and thus susceptible to over-forgetting. However, existing unlearning datasets lack this feature, as they are primarily focused on general unlearning tasks.
>
> For our study, we selected TOFU as the most appropriate dataset for our experiments. Additionally, following the suggestion from Reviewer [gJPq](https://openreview.net/forum?id=N2wPtFVK6o&noteId=IjJoHmgQ8q), **we incorporated another dataset, MUSE, which comprises both book and news corpora**. Our experiments with this additional dataset yielded results consistent with those from the TOFU dataset. **Please refer to Tables A1 and A2 in the global responses for detailed results and analysis, as well as Tables B1, B2, and B3 for the results of one additional baseline**.

---

> ### Author Response · Authors · 2024-11-25
> **Gentle Reminder**
>
> Dear Reviewer bVJx,
>
> Thank you for your time and effort in reviewing our paper! Since the discussion period will conclude in a few days, we would greatly appreciate it if you could review our responses and share any additional feedback or thoughts before the deadline (on November 26th AoE).
>
> Best regards,
>
> Submission 7011 Authors

---

> ### Author Response · Authors · 2024-12-02
> **Follow-up on Reviewer Feedback**
>
> Dear Reviewer bVJx,
>
> Thank you for your time and thoughtful feedback on our submission. As we approach the end of the discussion period (with approximately one day remaining), we noticed that you have not yet responded to our responses to your concerns.
>
> We believe we have carefully addressed all your comments and made every effort to enhance the quality of our work based on your valuable suggestions. These include clarifying the advantages of using ChatGPT in the current context and providing extended experimental results on two additional datasets.
>
> Given these updates, we kindly ask if you would be willing to review our rebuttal. If there are any remaining concerns or areas requiring further clarification, we would greatly appreciate your feedback before the discussion period concludes.
>
> Thank you once again for your time and consideration.
>
> Best regards,
>
> Submission 7011 Authors

---

### Author Response · Authors · 2024-11-20
**Additional Experiment Results (2/2)**

**Table A2.** MOUCHI's Performance on the MUSE-News Dataset
| \\( \mathcal{L}_{fgt} \\) | \\( \mathcal{L}_{rt} \\) | MOUCHI | Forget ↓        | Derivative ↑     | Retain ↑        |
|:-------------------------:|:------------------------:|:--------:|:-----------------:|:------------------:|:-----------------:|
|  GA                   | none                   | ✗      | 0.000*±0.000    | 0.000*±0.000     | 0.000*±0.000    |
|                         |                        | ✓      | 0.000*±0.000    | **0.444±0.103**      | **0.312±0.093**     |
|                       | GD                     | ✗      | 0.043*±0.004    | 0.410±0.115      | 0.184±0.061     |
|                         |                        | ✓      | 0.047*±0.005    | **0.820±0.109**      | **0.561±0.092**     |
|                         | KL                     | ✗      | 0.000*±0.000    | 0.000*±0.000     | 0.000*±0.000    |
|                         |                        | ✓      | 0.080*±0.025    | **0.556±0.063**      | **0.396±0.108**     |
|  DPO                    | none                   | ✗      | 0.002*±0.021    | 0.017*±0.107     | 0.001*±0.002    |
|                         |                        | ✓      | **0.261±0.057**     | **0.970±0.031**      | **0.782±0.079**     |
|                         | GD                     | ✗      | **0.151±0.021**    | 0.831±0.044      | 0.654±0.104     |
|                         |                        | ✓      | 0.192±0.039     | **0.978±0.043**      | **0.765±0.081**     |
|                         | KL                     | ✗      | 0.010*±0.010    | 0.040*±0.035     | 0.021*±0.023    |
|                         |                        | ✓      | **0.278±0.060**     |**0.733±0.066**      | **0.639±0.057**     |
|  NPO                    | none                   | ✗      | 0.056±0.038     | 0.524±0.065      | 0.353±0.101     |
|                         |                        | ✓      | **0.138±0.027**     | **0.881±0.042**      | **0.576±0.086**    |
|                         | GD                     | ✗      | **0.152±0.040**     | 0.558±0.073      | 0.357±0.125     |
|                         |                        | ✓      | 0.214±0.029     | **0.893±0.091**      | **0.869±0.113**     |
|                         | KL                     | ✗      | 0.014*±0.002    | 0.061*±0.028     | 0.057*±0.036    |
|                         |                        | ✓      | **0.155±0.042**     | **0.874±0.056**      | **0.751±0.066**     |
| TV                      |                        | ✗      | 0.741±0.025     | 0.619±0.042      | 0.736±0.102     |
|                         |                        | ✓      | **0.721±0.016**     | **0.815±0.072**      | **0.741±0.009**     |
|                         |    |   **Best Improvement**     | **28.9%**       | **60%**          | **143.4%**      |

**Table B1.** TV on TOFU
| MOUCHI | Forget ↓     | Derivative ↑  | Retain ↑     | Normal ↑     |
|:--------:|:--------------:|:---------------:|:--------------:|:--------------:|
| ✗      | **0.762±0.011**  | 0.752±0.071   | 0.735±0.054  | **0.491±0.031**  |
| ✓      | 0.771±0.082  | **0.831±0.033**   | **0.761±0.078** | 0.482±0.082  |

**Table B2.** TV on MUSE-Books
| MOUCHI | Forget ↓     | Derivative ↑  | Retain ↑     |
|:--------:|:--------------:|:---------------:|:--------------:|
| ✗      | **0.831±0.072**  | 0.852±0.033   | 0.811±0.041  |
| ✓      | 0.850±0.017  | **0.903±0.031**   | **0.866±0.053**  |

**Table B3.** TV on MUSE-News
| MOUCHI | Forget ↓     | Derivative ↑  | Retain ↑     |
|:--------:|:--------------:|:---------------:|:--------------:|
| ✗      | 0.741±0.025  | 0.619±0.042   | 0.736±0.102  |
| ✓      | **0.721±0.016**  | **0.815±0.072**   | **0.741±0.009**  |

---

### Author Response · Authors · 2024-11-20
**Additional Experiment Results (1/2)**

**Table A1.** MOUCHI's Performance on the MUSE-Books Dataset
| \\( \mathcal{L}_{fgt} \\) | \\( \mathcal{L}_{rt} \\) | MOUCHI | Forget ↓       | Derivative ↑    | Retain ↑        |
|:-------------------------:|:------------------------:|:--------:|:----------------:|:-----------------:|:-----------------:|
|  GA                  | none                       | ✗      | 0.468±0.154    | 0.096*±0.075    | 0.034*±0.013    |
|                         |                        | ✓      | **0.383±0.074**| **0.723±0.199** | **0.844±0.107** |
|                       | GD                     | ✗      | 0.648±0.133    | 0.913±0.034*    | 0.924±0.066     |
|                         |                        | ✓      | **0.542±0.056**| **0.970±0.012** | 0.924±0.031     |
|                         | KL                     | ✗      | 0.623±0.089    | 0.952±0.021     | 0.942±0.070     |
|                         |                        | ✓      | **0.601±0.045**| **0.996±0.006** | **0.988±0.014** |
|  DPO                    | none                   | ✗      | 0.012*±0.011   | 0.035*±0.034    | 0.332±0.112     |
|                         |                        | ✓      | **0.196±0.087**| **0.990±0.017** | 0.429±0.118     |
|                         | GD                     | ✗      | 0.213±0.088    | 0.853±0.040     | 0.488±0.073     |
|                         |                        | ✓      | **0.182±0.038**| **0.970±0.072** | 0.225±0.092     |
|                         | KL                     | ✗      | 0.032*±0.128   | 0.902±0.052     | 0.905±0.064     |
|                         |                        | ✓      | **0.511±0.081**| **0.991±0.010** | **0.938±0.026** |
|  NPO                    | none                   | ✗      | 0.444±0.170    | 0.718±0.192     | 0.835±0.100     |
|                         |                        | ✓      | **0.375±0.153**| **0.993±0.008** | **0.981±0.023** |
|                         | GD                     | ✗      | 0.480±0.163    | 0.814±0.108     | 0.864±0.093     |
|                         |                        | ✓      | **0.496±0.159**| **0.993±0.008** | **0.947±0.046** |
|                         | KL                     | ✗      | 0.472±0.081    | 0.512±0.084     | 0.594±0.013     |
|                         |                        | ✓      | **0.419±0.115**| **0.997±0.004** | **0.997±0.004** |
| TV                      |                        | ✗      | 0.831±0.072    | 0.852±0.033     | 0.811±0.041     |
|                         |                        | ✓      | **0.850±0.017**| **0.903±0.031** | **0.866±0.053** |
|                         |    |   **Best Improvement**     | **14.5%**      | **94.7%**       | **66.3%**       |

---

### Author Response · Authors · 2024-11-20
**Response Summary to All Reviewers**

We profoundly appreciate the reviewers' valuable feedback and constructive comments, which have greatly contributed to improving our paper. Overall, the reviewers have recognized the novelty of analyzing and identifying the over-forgetting problem (**all reviewers**), the clarity of the paper (**all reviewers**), and the usability of MOUCHI with existing approaches (Reviewers **[bVJx](https://openreview.net/forum?id=N2wPtFVK6o&noteId=FIWnvNrZnY)** and **[w5RU](https://openreview.net/forum?id=N2wPtFVK6o&noteId=9b9zKLG1IT)**). However, the reviewers have suggested that additional comparisons with recent datasets and baselines are necessary to further support the claim of MOUCHI's effectiveness in mitigating over-forgetting. We believe that the following responses address all the issues and questions raised by the reviewers.

### **Additional Experiments**
- According to the reviewers' suggestions, we have included a more recent dataset, MUSE [1], incorporated the MMLU benchmark [2] for general utility evaluation, and added a recent baseline, Task Vector (TV) [3,4].

### **Further Clarifications of MOUCHI's Design Choices**
- The main reason for initially focusing on a single dataset was the absence of datasets specifically designed for removing copyrighted information; existing ones target general unlearning tasks. We selected TOFU [5] as it was the closest match to our research objectives, though it still required some modifications. Based on the reviewers' suggestions, we expanded our experiments to include the MUSE dataset, which we also found to be suitable for our study.
- The robustness of ChatGPT as a substitute for experts in obtaining boundary KL divergence values was tested by generating the boundary QnA multiple time and averaging the results.


Once again, we sincerely thank the reviewers for their time and thoughtful suggestions to improve our paper. We are happy to address any additional questions during the discussion period. With these clarifications, we hope the reviewers will now feel confident in supporting this paper for publication by reconsidering their scores.

---

**References**

[1] Weijia Shi *et al*. Muse: Machine unlearning six-way evaluation for language models. *arXiv preprint arXiv:2407.06460, 2024*.

[2] Dan Hendrycks *et al*. In *Proceedings of the 9th International Conference on Learning Representations (ICLR)*, 2021.

[3] Gabriel Ilharco *et al*. Editing models with task arithmetic. In *Proceedings of the 11th International Conference on Learning
Representations (ICLR)*, 2023

[4] Guangyao Dou *et al*. Avoiding copyright infringement via machine unlearning. *arXiv preprint arXiv:2406.10952*, 2024.

[5] Pratyush Maini *et al*. TOFU: A task of fictitious unlearning for LLMs. *In Proceedings of the Conference on Language Modeling (COLM)*, 2024.

---

### Author Response · Authors · 2024-11-20
**Revision Note**

Based on the reviewers' feedback, we have highlighted the revised sections of the paper in **red**. The updates include:
- Improving the consistency of the derivative knowledge's definition (Reviewer **[gJPq](https://openreview.net/forum?id=N2wPtFVK6o&noteId=IjJoHmgQ8q)**)
- Adding additional related works (Reviewer **[c6sN](https://openreview.net/forum?id=N2wPtFVK6o&noteId=k2029RKIGj)**)
- Including the MMLU benchmark for broader utility evaluation (Reviewer **[c6sN](https://openreview.net/forum?id=N2wPtFVK6o&noteId=k2029RKIGj)**)
- Adding more recent baselines and datasets (**all reviewers**)
- Modifying the experiment section to reflect the addition of new baselines and datasets
- Updating the qualitative result of MOUCHI for better clarity (Reviewers **[w5RU](https://openreview.net/forum?id=N2wPtFVK6o&noteId=9b9zKLG1IT)** and **[gJPq](https://openreview.net/forum?id=N2wPtFVK6o&noteId=IjJoHmgQ8q)**)

---

### Meta-Review · Area_Chair_TMKh · 2024-12-20

**Metareview:**

This paper addresses the issue of over-forgetting during the unlearning process of Large Language Models (LLMs), where similar content that should be preserved is unintentionally removed. The authors introduce the concept of derivative knowledge, a subset of information derived from copyrighted content that must be retained during unlearning. Specifically, 1) the set of derivative knowledge is generated, and 2) a derivative loss function is proposed. The suggested method can be incorporated into various unlearning approaches, leading to improvements in both model utility and the quality of forgetting. However, most reviewers argue that the proposed method lacks technical contribution, as it simply prompts the model to generate derivative knowledge and uses KL divergence for filtering. Additionally, there is no improvement in the loss function for the derivative set, which seems to be an extension of the retain set. Besides, there are missing details regarding the implementation.

**Additional Comments On Reviewer Discussion:**

The reviewers had discussions with the authors, but most reviewers believe the paper lacks innovation.

---

### Decision · Program_Chairs · 2025-01-22

Reject